# UniCon: Unified Framework for Efficient Contrastive Alignment via Kernels

**Hangke Sui** [1,3*]**, Yuqing Wang**[2,3*]**, Minh N. Do**[1,2,3,4]

* Equal Contribution.

[1] Department of Electrical & Computer Engineering, The Grainger College of Engineering, UIUC

[2] Siebel School of Computing and Data Science, The Grainger College of Engineering, UIUC

[3] Coordinated Science Laboratory, University of Illinois Urbana-Champaign (UIUC)

[4] VinUni-Illinois Smart Health Center

{hangkes2, yuqing14, minhdo}@illinois.edu

## ABSTRACT

Contrastive objectives power state-of-the-art multimodal models, but their training remains slow, relying on long stochastic optimization. We propose a Unified Framework for Efficient Contrastive Alignment via Kernels (UniCon), which spans linear and nonlinear encoders as well as one-to-one and many-to-many alignments. At its core, UniCon introduces the contrastive similarity weight matrix $S(\gamma)$, which enables closed-form global solutions that provably replace minibatch backpropagation with exact updates. Through the lens of reproducing kernel Hilbert spaces (RKHS), UniCon provides a kernelized perspective that unifies contrastive alignment and reveals its connection to spectral methods. To validate the theory, we conduct experiments on synthetic, unimodal, multimodal, and zero-shot tasks, demonstrating that UniCon achieves substantial efficiency gains while preserving generality and strong empirical performance.

## 1 INTRODUCTION

Learning semantically aligned representations across different modalities, such as vision and language, has long been a central goal in machine learning (Ngiam et al., 2011; Srivastava & Salakhutdinov, 2012). In particular, *Multimodal Contrastive Learning* (MMCL)(Huang et al., 2024) has recently achieved remarkable success in zero-shot classification (Radford et al., 2021; Jia et al., 2021), cross-modal retrieval (Mu et al., 2022; Goel et al., 2022), and general visual understanding (Surís et al., 2023; Lin et al., 2023). These models typically train modality-specific encoders, e.g., a vision encoder and a language encoder, such that paired inputs are mapped to nearby representations in a shared space, while unpaired inputs are mapped far apart. At the heart of MMCL lies contrastive representation learning (Chopra et al., 2005; Gutmann & Hyvärinen, 2010; Sohn, 2016; Oord et al., 2018; Chen et al., 2020; Radford et al., 2021). Its versatility has made it a core component across diverse domains, including NLP (Gao et al., 2022; Izacard et al., 2021), bioimaging (Sanchez-Fernandez et al., 2023; Taleb et al., 2022; Han et al., 2022), recommendation (Xie et al., 2022; Yu et al., 2023; Jing et al., 2023; Yang et al., 2023), and graph learning (Kipf et al., 2019; You et al., 2020). The typical pipeline involves feature extraction via deep encoders and the optimization of contrastive loss.

Despite the impressive empirical performance of contrastive learning across vision, language, and multimodal domains, the theoretical foundations underlying its success remain only partially understood. There are works on the analysis of loss and training dynamics(Wang & Liu, 2021; Tian, 2022; HaoChen & Ma, 2022), provably guarantee of the model generalization (HaoChen et al., 2021; 2022; Tosh et al., 2021; Parulekar et al., 2023), duality between contrastive and non-contrastive method(Tian et al., 2021; Balestriero & LeCun, 2022). A growing body of theoretical work has sought to formalize contrastive learning (Saunshi et al., 2019; Tian et al., 2021; Jing et al., 2021; Wen & Li, 2021), often by simplifying the problem to single-modality settings. Recent advancements in contrastive learning have introduced novel loss functions and analytical frameworks to enhance representation quality and training efficiency.(Xu et al., 2023; Wang et al., 2024; Schuhmann et al., 2022). Analytical studies have examined contrastive learning from different perspectives. For example, Shi et al.

(2024) interpret the CLIP loss through the lens of optimal transport; while Tian (2022); Nakada et al. (2023) analyze multimodal contrastive learning using SVD- and PCA-based formulations, showing that, under *linear encoders*, contrastive loss minimization reduces to calculating a weighted covariance matrix. Yet, this insight has not been translated into nonlinear encoder settings and practical implementations.

We introduce **UniCon**, a *Unified Framework for Efficient Contrastive Alignment via Kernels*, which leverages a structured contrastive similarity weight matrix $S(\gamma)$ to directly solve contrastive objectives. As illustrated in Figure 1, UniCon departs from gradient-based training and instead: (i) in the linear setting(Nakada et al., 2023), solves a single spectral decomposition yielding optimal encoder matrices in closed form (ii) in a general nonlinear setting, provides a unified kernelized framework that enables fast alignment via implicit representation inference.

Our key contributions are as follows:

- Theoretically, we provide a kernel-based perspective that unifies linear and nonlinear encoders, showing that minimizing contrastive loss is equivalent to a spectral update. This leads to a provably optimal solution in closed form and connects contrastive learning to spectral methods.

- Beyond one-to-one matching, our framework generalizes to *many-to-many* alignment, broadening the applicability of contrastive alignment.

- Empirically, we demonstrate that UniCon converges fast and achieves competitive or superior performance across **synthetic**, **unimodal** (CIFAR-10), and **multimodal** (Flickr30k,MSCOCO) and **zero-shot transfer** (image-text retrieval), offering up to 461× speed-up over minimizing CLIP loss by stochastic gradient descent.

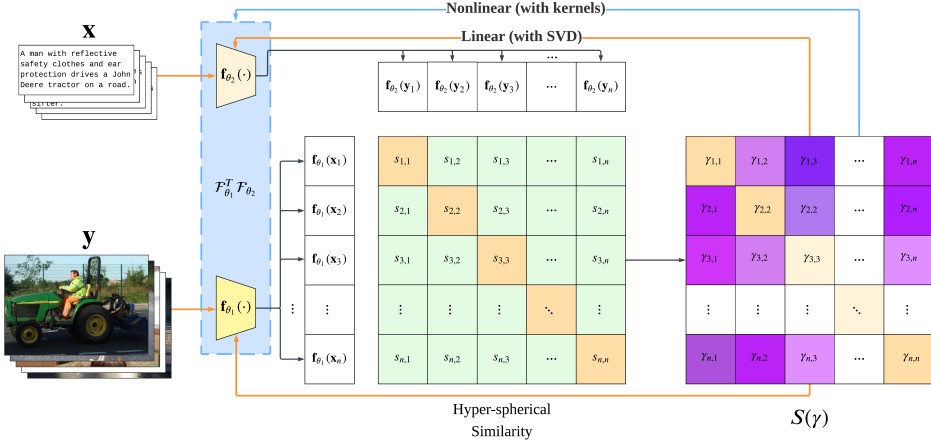

Figure 1: **Unified Framework for Efficient Contrastive Alignment via Kernels (UniCon)**. Starting from paired inputs, UniCon builds a contrastive similarity weight matrix $S(\gamma)$ using hyper-spherical similarities, then computes either (i) a closed-form spectral update in the linear case (orange) or (ii) a kernelized solution in the nonlinear case (blue).

## 2 BACKGROUND

**Contrastive Representation.** Contrastive learning (Chopra et al., 2005; Gutmann & Hyvärinen, 2010; Sohn, 2016; Oord et al., 2018; Chen et al., 2020; Radford et al., 2021) leverages paired inputs as a form of supervision. The central goal is to learn a representation space where *positive* (matching) pairs are mapped to nearby embeddings, while *negative* (non-matching) pairs are pushed apart. Learning representations on the hypersphere leads to better performance than in Euclidean space(Wang et al., 2017), as it avoids conflicting forces between attractive and repulsive gradients. (Wang & Isola, 2020) further shows that the distribution of representations on the unit hypersphere is encouraged to be uniform.

**Contrastive Loss** Contrastive loss (Hadsell et al., 2006) was first proposed in Siamese networks to pull together positive pairs and push apart negatives. The formulation was later unified under the InfoNCE loss (Oord et al., 2018), which serves as the basis for many self-supervised methods, including SimCLR (Chen et al., 2020). Supervised Contrastive (SupCon) loss (Khosla et al., 2020) extended contrastive learning to the supervised setting, where each anchor can have multiple positive samples from the same class. Building on the general family of contrastive losses introduced by (Tian, 2022), (Nakada et al., 2023) showed that these multimodal contrastive objectives can be connected to singular value decomposition (SVD) under linear setting.

**Kernel Method** Given a positive finite measure $\mu$ over a parameter space $\Theta$, we define a kernel $k : \mathcal{X} \times \mathcal{X} \to \mathbb{R}$ by $k(x, \tilde{x}) = \langle \phi(x; \theta), \phi(\tilde{x}; \theta) \rangle_\mu := \int_\Theta \phi(x; \theta) \phi(\tilde{x}; \theta) \, d\mu(\theta)$, which induces a Reproducing Kernel Hilbert Space (RKHS). Any function $f$ in this space admits the representation $f(x) = \sum_{j=1}^m w_j \, k(x, x_j), \; w_j \in \mathbb{R}$. Kernels are commonly used to learn representations (Kornblith et al., 2019; Klabunde et al., 2025), as they capture the relative structure among samples, critical for many learning algorithms (Aronszajn, 1950; Hofmann et al., 2008; Müller et al., 2018; Gong et al., 2025).

## 3 METHODOLOGY

We observe $N$ paired samples $\{(\mathbf{x}_i, \mathbf{y}_i)\}_{i=1}^N$, where $\mathbf{x}_i \in \mathbb{R}^{d_1}$ and $\mathbf{y}_i \in \mathbb{R}^{d_2}$. The objective of contrastive learning is to learn encoders $\mathbf{f}_{\theta_1} : \mathbb{R}^{d_1} \to \mathbb{R}^r$ and $\mathbf{f}_{\theta_2} : \mathbb{R}^{d_2} \to \mathbb{R}^r$ with modality-specific parameters $\theta_1$ and $\theta_2$, such that paired inputs are mapped to similar representations in a shared $r$-dimensional embedding space, while non-paired inputs remain dissimilar.

In the sections that follow, we first formalize the general contrastive learning framework, then analyze it under a linear representation setting, and finally unify our analysis spanning nonlinear encoders in RKHS. The proof can be found in the Appendix B.

**Definition 1** (Hyper-spherical similarity). *Define the similarity between $\mathbf{x}_i$ and $\mathbf{y}_i$ as the inner product on the hyper-sphere:*

$$s_{ij} = \langle \mathbf{f}_{\theta_1}(\mathbf{x}_i), \mathbf{f}_{\theta_2}(\mathbf{y}_j) \rangle_{\mathbb{S}^{r-1} \subset \mathbb{R}^r} = \left\langle \frac{\mathbf{f}_{\theta_1}(\mathbf{x}_i)}{\|\mathbf{f}_{\theta_1}(\mathbf{x}_i)\|_2}, \frac{\mathbf{f}_{\theta_2}(\mathbf{y}_j)}{\|\mathbf{f}_{\theta_2}(\mathbf{y}_j)\|_2} \right\rangle_{\mathbb{R}^r} = \frac{\mathbf{f}_{\theta_1}^\top(\mathbf{x}_i) \mathbf{f}_{\theta_2}(\mathbf{y}_j)}{\|\mathbf{f}_{\theta_1}(\mathbf{x}_i)\|_2 \|\mathbf{f}_{\theta_2}(\mathbf{y}_j)\|_2}. \quad (1)$$

**Definition 2** (Generalized contrastive loss). *Given $N$ paired samples $\{(\mathbf{x}_i, \mathbf{y}_i)\}_{i=1}^N$, we write the similarity matrix $[s_{ij}]$. With $\phi, \psi : \mathbb{R} \to \mathbb{R}_+$ monotonically increasing, scaling factor $\nu \geq 1$, and weights $\epsilon_{ij} \in [0, 1]$, the bidirectional general contrastive loss is*

$$L(\theta_1, \theta_2) = \frac{1}{2n} \sum_{i=1}^n \frac{1}{|\mathcal{P}_x(i)|} \sum_{k \in \mathcal{P}_x(i)} \phi\Big( \sum_{j \notin \mathcal{P}_x(i)} \epsilon_{ij} \psi(s_{ij} - \nu s_{ik}) + \epsilon_{ik} \psi(s_{ik} - \nu s_{ik}) \Big) \quad (2)$$

$$+ \frac{1}{2n} \sum_{i=1}^n \frac{1}{|\mathcal{P}_y(i)|} \sum_{k \in \mathcal{P}_y(i)} \phi\Big( \sum_{j \notin \mathcal{P}_y(i)} \epsilon_{ij} \psi(s_{ji} - \nu s_{ki}) + \epsilon_{ik} \psi(s_{ki} - \nu s_{ki}) \Big) + R(\theta_1, \theta_2)$$

*where $\mathcal{P}_x(i)$ denotes the index set of all samples in $\{y_i\}$ paired with $x_i$ while $\mathcal{P}_y(j)$ denotes the index set of all samples in $\{x_i\}$ paired with $y_j$. $|\mathcal{P}_x(i)|$ is the cardinality of the set $\mathcal{P}_x(i)$; and $R$ is an optional regularizer.*

This formulation naturally extends from *one-to-one* alignment (Tian, 2022) to *many-to-many* alignment: for example, a single image may correspond to multiple valid captions, and data augmentation can be viewed as creating diverse positive pairs. The scaling factor $\nu \geq 1$ adjusts the relative influence of positive pairs, while $\epsilon_{ij} \geq 0$ controls which pairs contribute to the loss (often $\epsilon_{ij} = 1$ for all negatives). The functions $\phi$ and $\psi$ are typically convex and monotonic, shaping the loss for optimization. By choosing specific forms for $\psi$ and $\phi$, we can recover familiar losses. More detailed examples can be found in Appendix A.

**Definition 3** (Contrastive similarity weight matrix). *Consider the general contrastive loss $\mathcal{L}(\theta_1, \theta_2)$, and a batch of paired samples $\{(\mathbf{x}_i, \mathbf{y}_i)\}_{i=1}^n$. Denote $\{\mathbf{e}_i\}_{i=1}^n$ as the elementary basis vectors of $\mathbb{R}^n$. The contrastive similarity weight is then defined as:*

$$S(\gamma) = -\frac{1}{n} \sum_{i,j} \frac{1}{2} \left( \frac{\gamma_{ij}}{|\mathcal{P}_x(i)|} + \frac{\bar{\gamma}_{ji}}{|\mathcal{P}_y(j)|} \right) \mathbf{e}_i \mathbf{e}_j^\top, \quad (3)$$

*with* weight coefficients

$$\gamma_{ij} = \begin{cases} \phi'_{ij} \cdot \left( \epsilon_{ij}(1-\nu)\psi'((1-\nu)s_{ij}) - \nu \sum_{m \notin P_x(i)} \epsilon_{im}\psi'(s_{im} - \nu s_{ij}) \right), & \text{if } j \in P_x(i) \\ \sum_{k \in P_x(i)} \phi'_{ik} \cdot (\epsilon_{ij}\psi'(s_{ij} - \nu s_{ik})), & \text{if } j \notin P_x(i) \end{cases} \quad (4)$$

$$\bar{\gamma}_{ij} = \begin{cases} \bar{\phi}'_{ij} \cdot \left( \epsilon_{ji}(1-\nu)\psi'((1-\nu)s_{ji}) - \nu \sum_{m \notin P_y(i)} \epsilon_{mi}\psi'(s_{mi} - \nu s_{ji}) \right), & \text{if } j \in P_y(i) \\ \sum_{k \in P_y(i)} \bar{\phi}'_{ik} \cdot (\epsilon_{ji}\psi'(s_{ji} - \nu s_{ki})), & \text{if } j \notin P_y(i) \end{cases} \quad (5)$$

*where*

$$\phi'_{ij} = \phi'(\epsilon_{ij}\psi((1-\nu)s_{ij}) + \sum_{m \notin P_x(i)} \epsilon_{im}\psi(s_{im} - \nu s_{ij})), \quad (6)$$

$$\bar{\phi}'_{ij} = \phi'(\epsilon_{ji}\psi(1-\nu)s_{ji} + \sum_{m \notin P_y(i)} \epsilon_{mi}\psi(s_{mi} - \nu s_{ji}))$$

The contrastive similarity weight matrix $S(\gamma)$ weighs pairwise interactions. We now show that minimizing the contrastive loss is equivalent to maximizing a new objective function with the constructed $S(\gamma)$.

**Lemma 4** (Gradient equivalence). *Consider minimizing the general contrastive loss (see Equation 2), the gradient of the contrastive loss with respect to encoder parameters satisfies:*

$$\frac{\partial \mathcal{L}}{\partial \theta_k} = - \left. \frac{\partial \operatorname{tr}\left( \mathcal{F}_{\theta_1}(\mathbf{X}) S(\gamma) \mathcal{F}_{\theta_2}^\top(\mathbf{Y}) \right)}{\partial \theta_k} \right|_{\gamma = \gamma(\theta_1, \theta_2)} + \frac{\partial R(\theta_1, \theta_2)}{\partial \theta_k}, \quad k \in \{1, 2\} \quad (7)$$

*where*

$$\mathbf{X} = [\mathbf{x}_1, \cdots, \mathbf{x}_n] \in \mathbb{R}^{d_1 \times n}, \mathbf{Y} = [\mathbf{y}_1, \cdots, \mathbf{y}_n] \in \mathbb{R}^{d_2 \times n}$$
$$\mathcal{F}_{\theta_1}(\mathbf{X}) = [\mathbf{f}_{\theta_1}(\mathbf{x}_1) \cdots \mathbf{f}_{\theta_1}(\mathbf{x}_n)] \in \mathbb{R}^{r \times n}, \mathcal{F}_{\theta_2}(\mathbf{Y}) = [\mathbf{f}_{\theta_2}(\mathbf{y}_1) \cdots \mathbf{f}_{\theta_2}(\mathbf{y}_n)] \in \mathbb{R}^{r \times n}.$$

This result reveals that the gradient of the contrastive loss in Equation 2 is the *negative* of the gradient of the proposed objective function in Equation 8. Hence, minimizing the contrastive loss is equivalent to maximizing the objective with contrastive similarity weight matrix $S(\gamma)$:

$$\operatorname{tr}\left( \mathcal{F}_{\theta_1}(\mathbf{X}) S(\gamma) \mathcal{F}_{\theta_2}^\top(\mathbf{Y}) \right) - R(\theta_1, \theta_2). \quad (8)$$

### 3.1 LINEAR REPRESENTATION SETTING

We first specialize the general framework to the linear representation case. Here the encoders are parameterized as matrix multiplications: $\mathbf{f}_{\theta_1}(\mathbf{x}) = F_1\mathbf{x}$, $\mathbf{f}_{\theta_2}(\mathbf{y}) = F_2\mathbf{y}$, where $F_1 \in \mathbb{R}^{r \times d_1}$, $F_2 \in \mathbb{R}^{r \times d_2}$ are learnable projection matrices.

**Definition 5** (Weighted contrastive covariance). *Define the weighted contrastive covariance as:*

$$C(\gamma) = \mathbf{X}S(\gamma)\mathbf{Y}^\top = -\frac{1}{n}\sum_{i,j} \frac{1}{2}\left( \frac{\gamma_{ij}}{|\mathcal{P}_x(i)|} + \frac{\bar{\gamma}_{ji}}{|\mathcal{P}_y(j)|} \right) \mathbf{x}_i\mathbf{y}_j^\top \quad (9)$$

*with coefficients $\gamma_{ij}$ share the same definition with Definition 3.*

**Remark 6.** *Note that the definition of $C(\gamma)$ exactly matches the definition of $S(\beta)$ in Nakada et al. (2023) under one-to-one alignment setting, which is proved in details in Appendix B.1. The structure of $C(\gamma)$ captures positive and negative pairs relationships, weighted appropriately. Our expression keeps the diagonal correction $(\alpha_{ii} + \bar{\alpha}_{ii})/2$ that prior work reduced to 1 by assuming that $\phi$ and $\psi$ are identity functions. This modification improves both theoretical generality and empirical performance.*

**Proposition 7.** *Under the linear setting, the Lemma 4 is specialized as*

$$\frac{\partial \mathcal{L}}{\partial F_k} = - \left. \frac{\partial \operatorname{tr}\left( F_1 C(\gamma) F_2^\top \right)}{\partial F_k} \right|_{\gamma = \gamma(F_1, F_2)} + \frac{\partial R(F_1, F_2)}{\partial F_k}, \quad k \in \{1, 2\} \quad (10)$$

To solve the optimization problem induced by our reformulated objective, we characterize its maximizer in the linear setting. We arrive at the following theorem, which establishes that the convergence of the contrastive loss can be replaced by a closed-form spectral update.

**Theorem 8** (Spectral characterization (Nakada et al., 2023)). *Consider minimizing the contrastive loss function* $\mathcal{L}(F_1, F_2)$, *with* $R(F_1, F_2) = \frac{\rho}{2}||F_1^T F_2||_F^2$. *Then,*

$$\underset{F_1, F_2}{\arg\min}\mathcal{L}(F_1, F_2) = \underset{F_1 \in \mathbb{R}^{r \times d_1}, F_2 \in \mathbb{R}^{r \times d_2}}{\arg\max} \operatorname{tr}\left(F_1 C(\gamma) F_2^\top\right) - \frac{\rho}{2}\left\|F_1^\top F_2\right\|_F^2 \qquad (11)$$

$$= \{(F_1, F_2) \in \mathbb{R}^{r \times d_1} \times \mathbb{R}^{r \times d_2} : F_1^\top F_2 = \frac{1}{\rho}\sum_{i=1}^r \sigma_i u_i v_i^\top\} \qquad (12)$$

*where* $\{\sigma_i, u_i, v_i\}_{i=1}^r$ *are the top-$r$ singular values and vectors of* $C(\gamma)$ *according to the Eckart–Young–Mirsky theorem.*

Consequently, in linear case, the global minimum is achieved by projecting the contrastive covariance $C(\gamma)$ onto its top-$r$ singular components. Thus, gradient descent on any loss in the contrastive family $(\phi, \psi)$ merely tracks the dominant singular subspace of $C(\gamma)$. Our algorithm UniCon performs this update in closed form, replacing thousands of SGD steps by one spectral factorization.

### 3.2 KERNELIZED REPRESENTATION SETTING

**Why leave the linear world?** The linear setting reveals a clean spectral structure for contrastive alignment, but cross–modal relations (e.g., vision $\leftrightarrow$ language) are typically nonlinear. Moreover, with frozen or partially frozen pretrained encoders, the residual alignment is rarely captured by mere linear heads. We therefore lift the analysis to **nonlinear** encoders while *keeping the output space* $\mathbb{R}^r$ *shared across modalities*. Kernelization provides a tractable route with explicit spectral solutions that *reduce to the linear case*.

**RKHS representation.** Let $(\mathcal{H}_X, k_X)$ and $(\mathcal{H}_Y, k_Y)$ be RKHSs with canonical feature maps

$$\phi_X(\mathbf{x}) = k_X(\cdot, \mathbf{x}) \in \mathcal{H}_X, \qquad \phi_Y(\mathbf{y}) = k_Y(\cdot, \mathbf{y}) \in \mathcal{H}_Y, \qquad (13)$$

satisfying the reproducing property $f(\mathbf{x}) = \langle f, \phi_X(\mathbf{x})\rangle_{\mathcal{H}_X}$ for all $f \in \mathcal{H}_X$ (and analogously for $\mathcal{H}_Y$). For $r$-dimensional outputs, the $a$-th coordinate $(a = 1, \ldots, r)$ admits the representer form

$$f_{\theta_1}^{(a)}(\cdot) = \sum_{i=1}^n A_{ia} \, k_X(\mathbf{x}_i, \cdot), \qquad f_{\theta_2}^{(a)}(\cdot) = \sum_{j=1}^n B_{ja} \, k_Y(\mathbf{y}_j, \cdot), \qquad (14)$$

with $A, B \in \mathbb{R}^{n \times r}$. Let $K_X = [k_X(\mathbf{x}_i, \mathbf{x}_j)]$ and $K_Y = [k_Y(\mathbf{y}_i, \mathbf{y}_j)]$. The batch embeddings are

$$\mathcal{F}_{\theta_1}(\mathbf{X}) = A^\top K_X \in \mathbb{R}^{r \times n}, \qquad \mathcal{F}_{\theta_2}(\mathbf{Y}) = B^\top K_Y \in \mathbb{R}^{r \times n}. \qquad (15)$$

The contrastive trace term becomes

$$\operatorname{tr}\!\left(\mathcal{F}_{\theta_1}(\mathbf{X})\, S(\gamma)\, \mathcal{F}_{\theta_2}(\mathbf{Y})^\top\right) = \operatorname{tr}\!\left(A^\top K_X\, S(\gamma)\, K_Y B\right). \qquad (16)$$

With the RKHS parameterization above and the kernelized trace form in equation 16, the entire objective can be written purely in terms of the Gram matrices. Under this notation, the optimizer is governed by the principal singular structure of $M := K_X^{1/2} S(\gamma) K_Y^{1/2}$, as formalized below.

**Theorem 9** (Kernelized spectral characterization (unified form)). *Let* $\rho > 0$ *and define the regularizer* $R(A, B) = \left\|(K_X^{1/2}A)^\top(K_Y^{1/2}B)\right\|_F^2$. *Then minimizing the contrastive loss is equivalent to the kernelized maximization*

$$\max_{A, B \in \mathbb{R}^{n \times r}} \operatorname{tr}\!\left(A^\top K_X\, S(\gamma)\, K_Y B\right) - \frac{\rho}{2}\left\|(K_X^{1/2}A)^\top(K_Y^{1/2}B)\right\|_F^2. \qquad (17)$$

*Let* $A' := A^\top K_X^{1/2}, B' := B^\top K_Y^{1/2}, M := K_X^{1/2} S(\gamma) K_Y^{1/2}$. *Then equation 17 rewrites*

$$\max_{A', B'} \operatorname{tr}(A' M B'^\top) - \frac{\rho}{2}\|A'^\top B'\|_F^2. \qquad (18)$$

*If* $M = U\Sigma V^\top$ *is an SVD and* $M_r = \sum_{i=1}^r \sigma_i u_i v_i^\top$ *its best rank-$r$ approximation by Eckart–Young–Mirsky theorem, then all maximizers satisfy the relation*

$$(A')^\top B' = \frac{1}{\rho} M_r \quad \Longleftrightarrow \quad A B^\top = \frac{1}{\rho} K_X^{-1/2} M_r K_Y^{-1/2}. \qquad (19)$$

*If* $K_X$ *or* $K_Y$ *is singular, replace inverse square roots by Moore–Penrose pseudo–inverse square roots.*

One explicit optimal choice is $A^\star = K_X^{-1/2} U_r$ and $B^\star = K_Y^{-1/2} V_r \Sigma_r / \rho$, where $U_r = [u_1, \ldots, u_r]$, $V_r = [v_1, \ldots, v_r]$, $\Sigma_r = \mathrm{diag}(\sigma_1, \ldots, \sigma_r)$ are SVD of $M$.

**Corollary 10** (Kernel inference (out-of-sample)). *Let $\mathcal{D} = \{(\mathbf{x}_i, \mathbf{y}_i)\}_{i=1}^n$ be the reference batch used to build contrastive similarity weight $S(\gamma)$ and let $k$ be a positive definite kernel. For a new pair $(\mathbf{x}^*, \mathbf{y}^*)$, set $\kappa_X(\mathbf{x}^*) = [k_X(\mathbf{x}_1, \mathbf{x}^*), \ldots, k_X(\mathbf{x}_n, \mathbf{x}^*)]^\top$ and $\kappa_Y(\mathbf{y}^*) = [k_Y(\mathbf{y}_1, \mathbf{y}^*), \ldots, k_Y(\mathbf{y}_n, \mathbf{y}^*)]^\top$. With an optimal $(A^\star, B^\star)$ from Theorem 9,*

$$\mathbf{f}_{\theta_1}(\mathbf{x}^*) = (A^\star)^\top \kappa_X(\mathbf{x}^*), \qquad \mathbf{f}_{\theta_2}(\mathbf{y}^*) = (B^\star)^\top \kappa_Y(\mathbf{y}^*), \tag{20}$$

*and the similarity*

$$s(\mathbf{x}^*, \mathbf{y}^*) = \frac{\kappa_X(\mathbf{x}^*)^\top A^\star B^{\star\top} \kappa_Y(\mathbf{y}^*)}{\|A^{\star\top} \kappa_X(\mathbf{x}^*)\|_2 \|B^{\star\top} \kappa_Y(\mathbf{y}^*)\|_2}. \tag{21}$$

In practice we observe that a simple angular kernel $k(u, v) = \frac{1}{\pi} \|u\| \|v\| \left( \sin \theta + (\pi - \theta) \cos \theta \right)$, $\theta = \arccos \left( \frac{u^\top v}{\|u\| \|v\|} \right)$ yields the best trade-off between speed and accuracy.

## 3.3 UNIFIED SPECTRAL VIEW (LINEAR AS A SPECIAL CASE OF KERNEL)

Our kernel formulation strictly generalizes the linear setting. For *linear kernels* $k_X(x, x') = \langle x, x' \rangle$, $k_Y(y, y') = \langle y, y' \rangle$, the Gram matrices reduce to $K_X = \mathbf{X}^\top \mathbf{X}$ and $K_Y = \mathbf{Y}^\top \mathbf{Y}$. Setting $F_1 = A^\top X^\top$ and $F_2 = B^\top Y^\top$ yields $\mathcal{R}_\times(A, B) = \|F_1^\top F_2\|_F^2 = \mathrm{tr}(F_1 F_1^\top F_2 F_2^\top)$, exactly matching the penalty used in the linear section. In the linear analysis we considered the weighted contrastive covariance matrix $C(\gamma) = X S(\gamma) Y^\top$, while in the RKHS analysis the central operator is $M = K_X^{1/2} S(\gamma) K_Y^{1/2}$. When the kernels are linear, $M = (X^\top X)^{1/2} S(\gamma) (Y^\top Y)^{1/2}$.

Let reduced SVDs be $X = U_X \Sigma_X V_X^\top$ and $Y = U_Y \Sigma_Y V_Y^\top$, and define $T = \Sigma_X V_X^\top S(\gamma) V_Y \Sigma_Y$ with SVD $T = U_T \Sigma_T V_T^\top$. Then we obtain

$$C(\gamma) = U_X T U_Y^\top = (U_X U_T) \Sigma_T (U_Y V_T)^\top, \qquad M = V_X T V_Y^\top = (V_X U_T) \Sigma_T (V_Y V_T)^\top. \tag{22}$$

Hence left/right multiplication by orthonormal matrices maps $C(\gamma)$ and $M$ to the *same* nonzero singular values $\Sigma_T$. In particular, their best rank–$r$ approximations select the same spectrum (Eckart–Young–Mirsky). When $X$ or $Y$ is rank-deficient (or for general kernels where $K_X$, $K_Y$ may be singular), all statements hold on the effective subspace using Moore–Penrose square roots.

Thus the kernel SVD of $M$ is the exact RKHS analogue of the linear SVD of $C(\gamma)$, and the linear setting is recovered as the special case of the linear kernel.

**Unified consequence.** Theorem 9 provides a unified spectral view for understanding contrastive alignment:

> contrastive loss minimization $\iff$ best rank–$r$ approximation with RKHS.

with $S(\gamma)$ encoding the particular contrastive objective (e.g., InfoNCE, CLIP, triplet) and the kernel selecting the nonlinear feature space in which the alignment is performed.

## 3.4 SCALABLE TRAINING

**Batch aggregation.** In practical scenarios, particularly in large-scale vision-language or multimodal applications, $N$ can be substantial, leading to prohibitive computational and memory demands. We therefore aggregate mini-batch contrastive similarity weight matrix $S^{(b)}(\gamma)$ by our closed-form solution. The final $S(\gamma)$ by taking quality weighted sum.

**Numerical stability.** If $K_X$ or $K_Y$ is ill-conditioned or singular, form square roots with a Tikhonov regularization when needed, replacing $K$ by $K + \lambda I$ with $\lambda > 0$ in $K^{\pm 1/2}$. This enhances robustness to near-singular Gram matrices and stabilizes the closed-form update. For low rank approximation step, one can use randomized SVD (Halko et al., 2011) on $M$ or Nyström approximations of $K_X$, $K_Y$ to reduce both memory and time.

# 4 EXPERIMENTS

During implementation, UniCon directly leverages the contrastive similarity weight $S(\gamma)$ for contrastive alignment. The full computation of $S(\gamma)$ is provided in Appendix C.1. We start with synthetic data and then extend to evaluate the practical utility of UniCon on unimodal and multimodal datasets. In the unimodal setting, we test on CIFAR-100, where the goal is image-to-image alignment. In the multimodal setting, we test on FLICKR30K and MS-COCO for image–text retrieval. Across both settings, we compare UniCon against standard CLIP-style contrastive learning trained with stochastic gradient descent (SGD–CLIP). All experiments are run on a single NVIDIA L40S GPU, and wall-clock times are reported for reference.

## 4.1 SYNTHETIC DATA

To verify our theoretical results in a fully controlled environment, we conduct synthetic experiments in both linear and nonlinear regimes. Details of setup could be found in AppendixC.2.

**Linear Latent-Factor Model.** We generate synthetic data from latent vectors $\mathbf{z} \in \mathbb{R}^r$ that are sampled around $K = 3$ cluster centers in latent space. We compare our method UniCon, which performs a spectral update using the closed-form SVD of the weighted covariance $C(\gamma)$, against a standard baseline trained via stochastic gradient descent (SGD) on the CLIP loss with AdamW optimizer ($\text{lr} = 2 \times 10^{-3}$). As shown in Figure 2, UniCon achieves $100\%$ matching accuracy after just 0.02 seconds. CLIP-SGD requires 400 epochs (0.32 seconds) to reach the same accuracy. This demonstrates that UniCon not only preserves structure in the latent space but also converges faster than gradient-based methods.

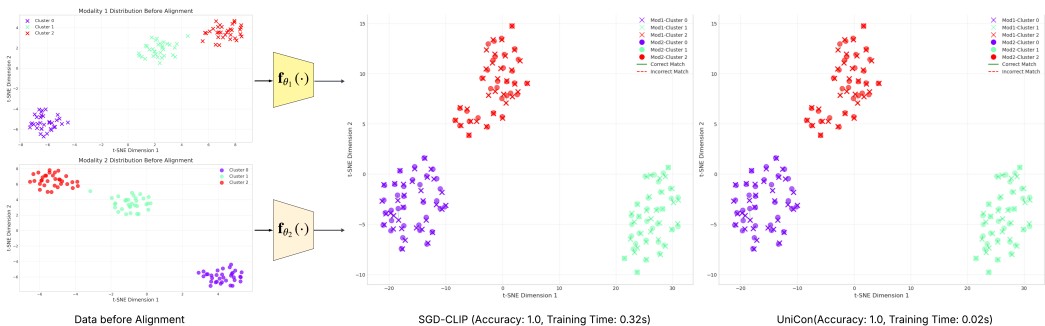

Figure 2: Visualization of cross-modal alignment using t-SNE embeddings of the shared representation space. Modality 1 (cross) and modality 2 (circle) are projected from different spaces into a shared representation space $\mathbb{R}^r$. Colors indicate ground-truth clusters, and lines connect matched image–text pairs. Both SGD-CLIP (left) and UniCon (right) successfully align paired samples while preserving cluster structure. The visual similarity between the two plots is expected: UniCon achieves a comparable aligned representation to SGD-CLIP with substantially less training time.

**Nonlinear Latent-Factor Model.** We further evaluate the method under a nonlinear transformation of the latent space. The baseline model trains nonlinear MLP encoder via SGD on CLIP loss and AdamW optimizer. Our method UniCon calculates a sequence of $\langle \mathcal{F}_{\theta_1}(\mathbf{x}) \mathcal{F}_{\theta_2}(\mathbf{y}) \rangle$ each for one training batch. Then we apply batch aggregation on validation data to calculate the weight for each training batch. The performance is evaluated by the correctly matched pairs of test data using the kernel-weighted generalization. UniCon converges in 2 epochs (0.04 seconds), achieving 86% matching accuracy, while CLIP-SGD reaches 84% after 500 epochs (0.65 seconds).

**Summary.** In both linear and nonlinear settings, UniCon demonstrates rapid convergence and strong alignment performance, validating the theoretical claims that contrastive learning objectives can be solved via a single spectral step. Figure 3 confirm that UniCon achieves consistent cross-modal alignment.

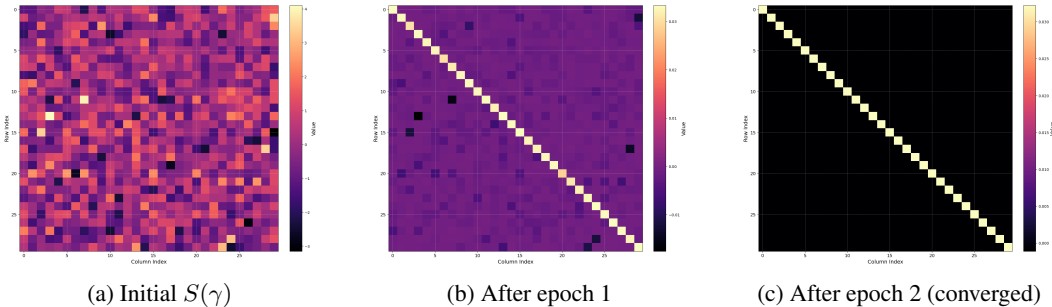

(a) Initial $S(\gamma)$        (b) After epoch 1        (c) After epoch 2 (converged)

Figure 3: Evolution of the contrastive similarity weight matrix $S(\gamma)$ in the nonlinear latent-factor model across training.

## 4.2 IMAGE-IMAGE ALIGNMENT ON CIFAR-10 (UNIMODAL)

**Setup.** We evaluate UniCon on a unimodal image alignment task using CIFAR-10, a benchmark dataset containing 10 object categories. Following the convention of SimCLR (Chen et al., 2020), we treat two augmentations derived from the same image as a positive pair, while augmentation pairs from different images are treated as negative. Feature embeddings are extracted using a frozen ResNet-18 encoder. The objective is to align these embeddings such that images from the same class are pulled closer together in the shared feature space, thereby facilitating unimodal classification.

We conduct this experiment under a nonlinear setting. For baseline, we train a lightweight projection encoder $g(\cdot)$ on frozen ResNet-18 features with bidirectional InfoNCE optimized by SGD. The encoder is a two-layer MLP encoder. We optimize with SGD for 300 epochs. The trained encoder is then frozen for linear probing with a small classifier on the 128-dimensional embeddings. UniCon learns a kernelized projection from frozen ResNet-18 features by a spectral closed-form solution. Given two augmented views $(z1, z2)$, we compute an angular kernel between features and iteratively estimate a batch-wise feature map $A$. After learning an average of $A$, we freeze it and train a small linear classifier on the 128-dimensional embeddings for linear probing, mirroring the SGD setup for fair comparison.

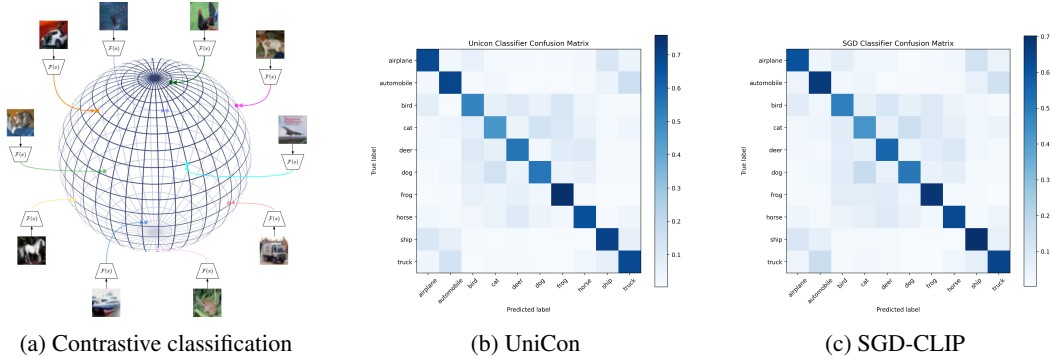

(a) Contrastive classification        (b) UniCon        (c) SGD-CLIP

Figure 4: **Visualizations of unimodal alignment on CIFAR-10.** (a) Self-supervised contrastive learning clusters semantically similar images and uniformly distributes clusters on the hypersphere. (b–c) Unimodal confusion matrices for UniCon and SGD-CLIP, showing predicted vs. true class accuracy. The near-identity structure and visual similarity of both matrices indicate that UniCon and SGD-CLIP achieve comparable discriminative performance in unimodal contrastive alignment.

**Results.** To evaluate classification performance, we report the confusion matrix in Figure 4(b-c), which summarizes the number of correct and incorrect predictions for each class. Each row of the matrix corresponds to the predicted class, while each column represents the ground truth. Specifically, the diagonal entries indicate the number of correctly classified samples, and the off-diagonal entries capture misclassifications between classes. This allows a detailed analysis of model behavior beyond overall accuracy. Numerically, UniCon can achieve the average accuracy of 61.82% with 23.38s while

SGD can achieve the average accuracy of 62.21% with 41.98s. Empirically, UniCon can converge within 2 epochs while SGD requires various iterations to converge to a comparable optimal point.

### 4.3 IMAGE–TEXT RETRIEVAL AND ZERO-SHOT TRANSFER (MULTIMODAL)

**Setup.** To evaluate UniCon in a multimodal setting, we benchmark on the standard image–text retrieval tasks from FLICKR30K and MSCOCO. We consider three backbone choices for image $x$ and text $y$: (a) **ResNet-18** (He et al., 2016) for images with **Sentence-BERT** (`all-mpnet-base-v2`) (Reimers & Gurevych, 2019) for text; (b) **ResNet-50** + Sentence-BERT; (c) the pretrained **CLIP ViT-B/32** model as a frozen visual–textual feature extractor. UniCon is compared against an SGD-optimized CLIP baseline (SGD–CLIP) under matched training/evaluation settings; both are trained to convergence.

**Results.** Table 1 summarizes top-1/10 recall in both directions. Across all backbones, UniCon attains competitive or superior accuracy while reducing training time by **25–50×**. With CLIP ViT-B/32 features, UniCon further improves accuracy despite requiring only a single spectral update. Notably, UniCon on Resnet50+SBERT backbone achieves comparable averaged top-10 retrieval accuracy with CLIP ViT-B/32 backbone aligned SGD-CLIP. These findings are consistent with our theory: the spectral step efficiently recovers the dominant cross-modal structure that iterative SGD approximates over many epochs.

Table 1: **Image-text retrieval on FLICKR30K.** We report Recall@1 and Recall@10 for both image→text and text→image directions.

| Backbone | Method | Train time | Image→Text | | Text→Image | | Average | |
|---|---|---|---|---|---|---|---|---|
| | | | R@1 | R@10 | R@1 | R@10 | R@1 | R@10 |
| RN-18 + SBERT | SGD–CLIP | 45.6 s | **.043** | **.221** | .041 | .217 | .042 | .219 |
| | **UniCon** | **1.7 s** | .020 | .145 | **.087** | **.361** | **.054** | **.253** |
| RN-50 + SBERT | SGD–CLIP | 45.0 s | .043 | .221 | .041 | .217 | .042 | .219 |
| | **UniCon** | **0.81 s** | **.134** | **.464** | **.188** | **.567** | **.161** | **.515** |
| CLIP ViT-B/32 | SGD–CLIP | 45.3 s | .231 | .595 | .241 | .600 | .236 | .597 |
| | **UniCon** | **0.76 s** | **.284** | **.636** | **.421** | **.777** | **.353** | **.701** |

Table 2: **Retrieval on MSCOCO and zero-shot transfer to FLICKR30K.** All models are trained on MSCOCO. We report image to text (I→T) and text to image (T→I) on MSCOCO and zero-shot on FLICKR30K (no fine-tuning).

| Backbone | Method | Train (s) | Dir. | MSCOCO | | FLICKR30K (zero-shot) | |
|---|---|---|---|---|---|---|---|
| | | | | R@1 | R@10 | R@5 | R@10 |
| RN-50 + SBERT | SGD–CLIP | 5121.72 | I→T | .053 | .253 | — | — |
| | | | T→I | .060 | .286 | | |
| | **UniCon** | **11.11** | I→T | **.105** | **.388** | .171 | .261 |
| | | | T→I | **.129** | **.439** | .249 | .353 |
| CLIP ViT-B/32 | SGD–CLIP | 1066.60 | I→T | .128 | .415 | — | — |
| | | | T→I | .123 | .427 | | |
| | **UniCon** | **11.15** | I→T | **.329** | **.685** | **.808** | **.879** |
| | | | T→I | **.292** | **.644** | **.766** | **.848** |

Table 2 augments our results with MSCOCO retrieval and *zero-shot* transfer to FLICKR30K. Our training follows the standard retrieval protocol on MSCOCO with each image paired with 5 captions, and report test retrieval accuracy on 5,000 held-out pairs. UniCon achieves higher accuracy than SGD–CLIP on MSCOCO while being **96–461×** faster. Beyond scalability, the learned alignment transfers robustly: models trained on MSCOCO maintain strong performance on FLICKR30K without any adaptation. Despite distribution shifts in both image and text domains, UniCon maintains

strong retrieval accuracy. These results underscore UniCon's scalability, generality, and cross-dataset transfer, reavealing its potential in real world tasks.

## 5 DISCUSSION

**Computation Efficiency.** UniCon achieves rapid stabilization of the alignment subspace through derived spectral updates, which bypass many small gradient steps, demonstrating computational efficiency. Details can be found in Appendix C.1. Empirically, we observe an interesting phenomenon that $M$ (or $C(\gamma)$ in the linear case) converges in 2 or a few steps. We provide an intuitive explanation: Unlike gradient-based methods that take small local steps, each spectral update directly jumps to the global maximizer of the surrogate objective, making the update much more informative.

**Data Efficiency.** Additionally, on MSCOCO, using only 200 images (0.24% of the dataset), with each image paired with 5 captions, already yields meaningful retrieval alignment (66.45% avg R@10), demonstrating both subspace convergence and data efficiency. As we discussed in Section 3, alignment is a r-rank discovery problem, which gives an intuition that we don't need massive datasets to find the principal axes.

**Static vs. Evolving Input Spaces.** The theoretical optimality results with r-rank aproximation is derived under the assumption that the input space of UniCon is static. It includes two cases: (a) Input space is data space (raw modalities), where UniCon itself performs end-to-end alignment. (b) Input space is embedding space from frozen encoders. In both cases, UniCon provides a globally optimal spectral solution to contrastive loss minimization from the perspective of r-rank approximation. When encoders are trainable (non-static input space), UniCon is applied during jointly optimizing encoders, the spectral update becomes a conditionally optimal subproblem, i.e., optimal for the current encoder outputs.

## 6 CONCLUSION

We propose **UniCon**, a theoretically grounded and computationally efficient framework for contrastive alignment that unifies linear and nonlinear encoders through kernels. We show that minimizing contrastive loss is equivalent to maximizing a kernelized trace objective, which in turn reduces to a best rank-r spectral approximation in an RKHS. The closed form update is driven by explicitly constructing a contrastive similarity weight matrix $S(\gamma)$. In the linear reduction, UniCon recovers a projection onto the top-r singular directions of the weighted contrastive cross-covariance. This yields a clear spectral lens on contrastive learning, interpreting alignment as r-rank structure discovery in high-dimensional feature spaces.

We see three concrete future directions: (i) structure-exploiting kernels, for example random features, to reduce K's rank and cost; (ii) hybrid spectral–SGD strategy or warm-start strategy when the input space is non-static (e.g. finetuning for domain adaptation);(iii) mini-batch aggregation and streaming variants that balance analytical fidelity with scalability, including distribution-shift-aware weighting, online updates of the aggregated operator, and robust selection of batches to maintain stable global estimates.

With the theoretical grounding and competitive empirical results, UniCon advances understanding of contrastive learning for unimodal representation learning, multimodal alignment and beyond.

### ACKNOWLEDGMENT

This work was supported in part by the Advanced Research Projects Agency for Health (ARPA-H) and the IBM–Illinois Discovery Accelerator Institute.

### ETHICS STATEMENT

We affirm that this work complies with the ICLR Code of Ethics. Our study does not involve human subjects, sensitive personal data, or potentially harmful applications. All datasets used are publicly

available (e.g., CIFAR-10) and contain no personally identifiable information. We acknowledge the importance of fairness and responsible AI development and have taken care to ensure that our methods do not propagate bias or cause unintended harm.

## REPRODUCIBILITY STATEMENT

We are committed to ensuring the reproducibility of our work. All implementation details, including model architectures, hyperparameters, and training procedures, are described in the Appendix C. We provide pseudocode for our algorithm in Appendix C and include complete proofs of theoretical results in Appendix B. All datasets used in this study are publicly available, and we will release source code and experiment scripts as supplementary materials later to facilitate replication of our results.

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

## Contents

## A   CONTRASTIVE LOSS

Under the assumption of one-to-one alignment, let $E_{\theta_1} : \mathcal{X} \to \mathbb{R}^r$ and $E_{\theta_2} : \mathcal{Y} \to \mathbb{R}^r$ denote two modality-specific encoders with trainable parameters $\theta_1, \theta_2$. Given $N$ paired samples $\{(\mathbf{x}_i, \mathbf{y}_i)\}_{i=1}^N$ we form the *similarity matrix $S = [s_{ij}]$*. All contrastive objectives used in practice can be written in the *bidirectional general form* (Tian, 2022; Nakada et al., 2023)

$$\mathcal{L}(\theta_1, \theta_2) = \frac{1}{2N} \sum_{i=1}^N \left[ \phi\left(\sum_{j=1}^N \epsilon_{ij}\, \psi\big(s_{ij} - \nu\, s_{ii}\big)\right) + \phi\left(\sum_{j=1}^N \epsilon_{ij}\, \psi\big(s_{ji} - \nu\, s_{ii}\big)\right) \right] + R(\theta_1, \theta_2), \quad (23)$$

with

- $\psi, \phi \colon \mathbb{R} \to \mathbb{R}$ increasing (shape of the loss),
- $\nu$ : relative weight on the positive pair,
- $\epsilon_{ij} \in [0, 1]$ : which pairs are used,
- $R$ : optional regulariser (e.g. weight decay).

**Remark** *Because the embeddings are length-normalised, $s_{ij} \in [-1, 1]$ and all geometry lives on the unit hypersphere.*

By choosing specific forms for $\psi$ and $\phi$, we can recover familiar losses. For example, choosing $\phi(x) = \tau \log(x)$, $\psi(x) = \exp(x/\tau)$ and including positive pairs in the normalization ($\epsilon_{ij} = 1$ for positive pair $(i, j)$), recovers the CLIP(Radford et al., 2021) loss, and the InfoNCE(Oord et al., 2018) loss is the same instantiation appears as a simplified variant focusing only on one direction. And choosing $\phi(x) = x$, $\psi(x) = [-x + \epsilon]_+$ gives triplet loss(Schroff et al., 2015). Equation 2 thus unifies a wide spectrum of contrastive objectives via variant choices of $(\phi, \psi, \nu, \epsilon)$, providing a common lens for analysing and extending multimodal representation learning. This formulation allows for distinct temperature scaling of positive and negative similarities.

**Derivation for CLIP (Radford et al., 2021) / InfoNCE (Oord et al., 2018) Loss.** Set $\psi(x) = e^{x/\tau}$, $\phi(x) = \tau \log x$, $\nu = 1$, $\epsilon_{ij} = 1 - \delta_{ij}$ in equation 23, and omit $R$. We define

$$
\delta_{ij} = \begin{cases} 1, & \text{if } i = j \\ 0, & \text{if } i \neq j \end{cases} \tag{24}
$$

This gives the loss

$$
\mathcal{L} \triangleq \frac{1}{2n} \sum_i \phi \left( \sum_{j \in [n]} \epsilon_{ij} \psi \left( s_{ij} - \nu s_{ii} \right) \right) + \frac{1}{2n} \sum_i \phi \left( \sum_{j \in [n]} \epsilon_{ij} \psi \left( s_{ji} - \nu s_{ii} \right) \right) + R \left( \theta_1, \theta_2 \right) \tag{25}
$$

$$
= \frac{\tau}{2n} \sum_i \log \left( \sum_{j \in [n]} \exp \left( \frac{s_{ij} - s_{ii}}{\tau} \right) \right) + \frac{\tau}{2n} \sum_i \log \left( \sum_{j \in [n]} \exp \left( \frac{s_{ji} - s_{ii}}{\tau} \right) \right) \tag{26}
$$

$$
= \frac{\tau}{2n} \sum_i \log \left( \frac{\sum_{j \in [n]} \exp \left( \frac{s_{ij}}{\tau} \right)}{\exp \left( \frac{s_{ii}}{\tau} \right)} \right) + \frac{\tau}{2n} \sum_i \log \left( \frac{\sum_{j \in [n]} \exp \left( \frac{s_{ji}}{\tau} \right)}{\exp \left( \frac{s_{ii}}{\tau} \right)} \right) \tag{27}
$$

$$
= \frac{\tau}{2n} \sum_i \left[ -\log \left( \frac{\exp \left( \frac{s_{ii}}{\tau} \right)}{\sum_{j \in [n]} \exp \left( \frac{s_{ij}}{\tau} \right)} \right) - \log \left( \frac{\exp \left( \frac{s_{ii}}{\tau} \right)}{\sum_{j \in [n]} \exp \left( \frac{s_{ji}}{\tau} \right)} \right) \right] \tag{28}
$$

$$
= \mathcal{L}_{CLIP} \tag{29}
$$

For InfoNCE loss, we keep the first term (i.e. only one-directional loss), then

$$
\mathcal{L}_{InfoNCE} = \frac{\tau}{n} \sum_i \left[ -\log \frac{\exp s_{ii}}{\sum_j \exp s_{ij}} \right]
$$

**Derivation for triplet loss (Schroff et al., 2015).** With a margin $\epsilon > 0$, choose $\psi(x) = [\epsilon - x]_+$, $\phi(x) = x$, $\nu = 1$, $\epsilon_{ij} = 1 - \delta_{ij}$,

$$
\mathcal{L} \triangleq \frac{1}{2n} \sum_i \phi \left( \sum_{j \in [n]} \epsilon_{ij} \psi \left( s_{ij} - \nu s_{ii} \right) \right) + \frac{1}{2n} \sum_i \phi \left( \sum_{j \in [n]} \epsilon_{ij} \psi \left( s_{ji} - \nu s_{ii} \right) \right) + R \left( \theta_1, \theta_2 \right) \tag{30}
$$

$$
= \frac{1}{2n} \sum_{i=1}^n \left[ \sum_{\substack{j=1 \\ j \neq i}}^n \left[ \epsilon - \left( s_{ij} - s_{ii} \right) \right]_+ + \sum_{\substack{j=1 \\ j \neq i}}^n \left[ \epsilon - \left( s_{ji} - s_{ii} \right) \right]_+ \right] \tag{31}
$$

We can also only keep one direction:

$$
\mathcal{L} = \frac{1}{n} \sum_{i=1}^n \left[ \sum_{\substack{j=1 \\ j \neq i}}^n \left[ \epsilon - \left( s_{ij} - s_{ii} \right) \right]_+ \right] = \frac{1}{n} \sum_{i=1}^n \left[ \sum_{\substack{j=1 \\ j \neq i}}^n \max\{0, s_{ii} - s_{ij} + \epsilon\} \right] \tag{32}
$$

where $s_{ij}$ captures distance between negative pairs, and $s_{ii}$ captures distance between positive pairs.

Equation equation 23 thus provides a general form that captures various contrastive loss, making it possible to analyze them collectively and design new variants with principled control over positive / negative balance, temperature, and weighting.

**Many-to-many aligment contrastive loss(Khosla et al., 2020)** Note that the loss in Equation equation 23 is defined with $(x_i, y_i)$ being the positive pairs for all $i$. However, in many cases, a single $x_i$ may have multiple positive pairs. Khosla et al. (2020) extended contrastive learning to the supervised setting with the Supervised Contrastive (SupCon) loss, which is not restricted to one-to-one pairs. This loss encourages embeddings from the same class to be pulled together while pushing apart

embeddings from different classes. Formally, given a minibatch of normalized embeddings $z_i$ with labels $y_i$, the SupCon loss is defined as

$$\mathcal{L}_{out}^{sup} = \sum_{i \in I} L_{out,i}^{sup} = \sum_{i \in I} \frac{-1}{|P(i)|} \sum_{p \in P(i)} \log \frac{\exp(\boldsymbol{z}_i \cdot \boldsymbol{z}_p / \tau)}{\sum_{a \in A(i)} \exp(\boldsymbol{z}_i \cdot \boldsymbol{z}_a / \tau)}. \tag{33}$$

where $P(i) \equiv \{p \in A(i) : \tilde{y}_p = \tilde{y}_i\}$ is the set of indices of all positives in the multiviewed batch distinct from $i$ and $|P(i)|$ is its cardinality.

Therefore, we extend the general form of the contrastive loss in equation 23 to handle many-to-many alignment scenarios. We define the unified form as

$$L(\theta_1, \theta_2) = \frac{1}{2n} \sum_{i=1}^{n} \frac{1}{|\mathcal{P}_x(i)|} \sum_{k \in \mathcal{P}_x(i)} \phi\Big( \sum_{j \notin \mathcal{P}_x(i)} \epsilon_{ij} \psi(s_{ij} - \nu s_{ik}) + \epsilon_{ik} \psi(s_{ik} - \nu s_{ik}) \Big)$$

$$+ \frac{1}{2n} \sum_{i=1}^{n} \frac{1}{|\mathcal{P}_y(i)|} \sum_{k \in \mathcal{P}_y(i)} \phi\Big( \sum_{j \notin \mathcal{P}_y(i)} \epsilon_{ij} \psi(s_{ji} - \nu s_{ki}) + \epsilon_{ik} \psi(s_{ki} - \nu s_{ki}) \Big) + R(\theta_1, \theta_2)$$

$$\tag{34}$$

where $\mathcal{P}_x(i)$ and $\mathcal{P}_y(j)$ denote the index sets of all samples in $\{y_k\}$, $\{x_k\}$ paired with $x_i$ and $y_j$, respectively. The term $|\mathcal{P}_x(i)|$ denotes the cardinality of $\mathcal{P}_x(i)$, and $R$ is an optional regularization term. We incorporate this loss function in the following sections.

## B  THEORETICAL PROOFS

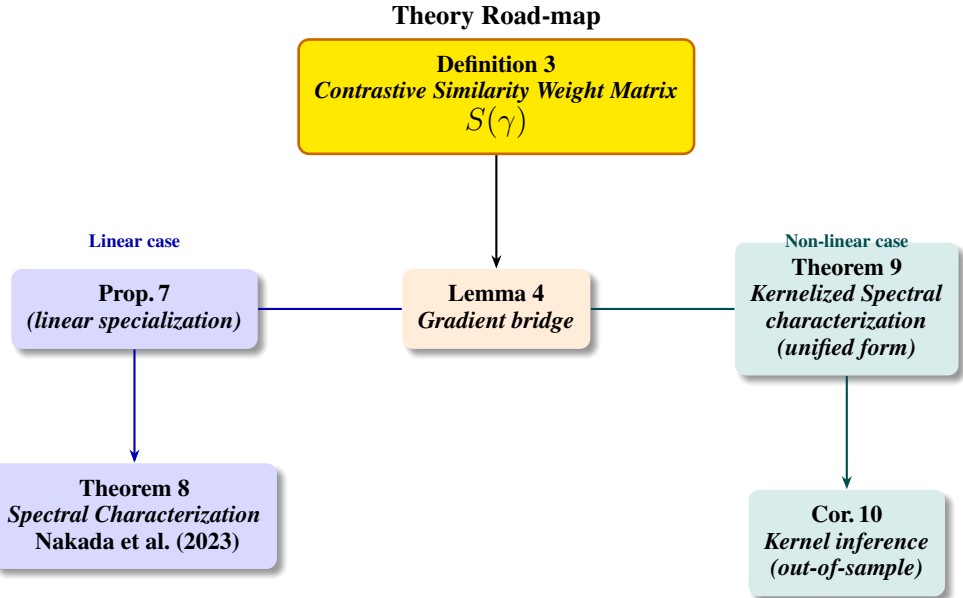

**Definition 3** (Contrastive similarity weight matrix) *Consider the general contrastive loss $\mathcal{L}(\theta_1, \theta_2)$ in Equation equation 34 with choice of $(\phi, \psi, \epsilon, \nu)$, and a batch of samples $\{(\mathbf{x}_i, \mathbf{y}_i)\}_{i=1}^n$. Denote $\{\mathbf{e}_i\}_{i=1}^n$ as the elementary basis vectors of $\mathbb{R}^n$. The contrastive similarity weight matrix is then defined as:*

$$S(\gamma) = -\frac{1}{n} \sum_{i,j} \frac{1}{2} \left( \frac{\gamma_{ij}}{|\mathcal{P}_x(i)|} + \frac{\bar{\gamma}_{ji}}{|\mathcal{P}_y(j)|} \right) \mathbf{e}_i \mathbf{e}_j^\top, \tag{35}$$

*with* weight coefficients

$$\gamma_{ij} = \begin{cases} \phi'_{ij} \cdot \Big( \epsilon_{ij}(1-\nu)\psi'((1-\nu)s_{ij}) - \nu \sum_{m \notin P_x(i)} \epsilon_{im} \psi'(s_{im} - \nu s_{ij}) \Big), & \text{if } j \in P_x(i) \\ \sum_{k \in P_x(i)} \phi'_{ik} \cdot (\epsilon_{ij} \psi'(s_{ij} - \nu s_{ik})), & \text{if } j \notin P_x(i) \end{cases} \tag{36}$$

$$\bar{\gamma}_{ij} = \begin{cases} \bar{\phi}'_{ij} \cdot \left( \epsilon_{ji}(1-\nu)\psi'((1-\nu)s_{ji}) - \nu \sum_{m \notin P_y(i)} \epsilon_{mi}\psi'(s_{mi} - \nu s_{ji}) \right), & \text{if } j \in P_y(i) \\ \sum_{k \in P_y(i)} \bar{\phi}'_{ik} \cdot (\epsilon_{ji}\psi'(s_{ji} - \nu s_{ki})), & \text{if } j \notin P_y(i) \end{cases} \tag{37}$$

*where we define*

$$\phi'_{ij} = \phi'\left( \epsilon_{ij}\psi((1-\nu)s_{ij}) + \sum_{m \notin P_x(i)} \epsilon_{im}\psi(s_{im} - \nu s_{ij}) \right), \tag{38}$$

$$\bar{\phi}'_{ij} = \phi'\left( \epsilon_{ji}\psi(1-\nu)s_{ji} + \sum_{m \notin P_y(i)} \epsilon_{mi}\psi(s_{mi} - \nu s_{ji}) \right) \tag{39}$$

**Lemma 4** (Gradient Equivalence) *Consider minimizing the general contrastive loss (see Equation equation 2), the gradient of the contrastive loss with respect to encoder parameters satisfies:*

$$\frac{\partial \mathcal{L}}{\partial \theta_k} = - \left. \frac{\partial \operatorname{tr}\left( \mathcal{F}_{\theta_1}(\mathbf{X}) S(\gamma) \mathcal{F}_{\theta_2}^\top(\mathbf{Y}) \right)}{\partial \theta_k} \right|_{\gamma = \gamma(\theta_1, \theta_2)} + \frac{\partial R(\theta_1, \theta_2)}{\partial \theta_k}, \quad k \in \{1, 2\} \tag{40}$$

*where*

$$\mathbf{X} = [\mathbf{x}_i, \cdots, \mathbf{x}_n] \in \mathbb{R}^{d_1 \times n}, \mathbf{Y} = [\mathbf{y}_i, \cdots, \mathbf{y}_n] \in \mathbb{R}^{d_2 \times n}$$
$$\mathcal{F}_{\theta_1}(\mathbf{X}) = [\mathbf{f}_{\theta_1}(\mathbf{x}_1)\ \mathbf{f}_{\theta_1}(\mathbf{x}_2)\ \cdots\ \mathbf{f}_{\theta_1}(\mathbf{x}_n)] \in \mathbb{R}^{r \times n}$$
$$\mathcal{F}_{\theta_2}(\mathbf{Y}) = [\mathbf{f}_{\theta_2}(\mathbf{y}_1)\ \mathbf{f}_{\theta_2}(\mathbf{y}_2)\ \cdots\ \mathbf{f}_{\theta_2}(\mathbf{y}_n)] \in \mathbb{R}^{r \times n}.$$

*Proof.* Let $\theta_{k,\ell}$ be the $\ell$-th component of $\theta_k$. We have

$$\partial_{\theta_{k,\ell}} \mathcal{L} = \partial_{\theta_{k,\ell}} \left[ \frac{1}{2n} \sum_i \frac{1}{|P_x(i)|} \sum_{k \in P_x(i)} \phi\left( \epsilon_{ik}\psi((1-\nu)s_{ik}) + \sum_{m \notin P_x(i)} \epsilon_{im}\psi(s_{im} - \nu s_{ik}) \right) \right.$$
$$\left. + \frac{1}{2n} \sum_i \frac{1}{|P_y(i)|} \sum_{k \in P_y(i)} \phi\left( \epsilon_{ki}\psi((1-\nu)s_{ki}) + \sum_{m \notin P_y(i)} \epsilon_{mi}\psi(s_{mi} - \nu s_{ki}) \right) + R(\theta_1, \theta_2) \right] \tag{41}$$

$$= \frac{1}{2n} \sum_{i=1}^n \frac{1}{|P_x(i)|} \sum_{k \in P_x(i)} \phi'\left( \epsilon_{ik}\psi((1-\nu)s_{ik}) + \sum_{m \notin P_x(i)} \epsilon_{im}\psi(s_{im} - \nu s_{ik}) \right)$$
$$\cdot \left[ \epsilon_{ik}\psi'((1-\nu)s_{ik})(1-\nu)\partial_{\theta_{k,\ell}} s_{ik} + \sum_{m \notin P_x(i)} \epsilon_{im}\psi'(s_{im} - \nu s_{ik})\left( \partial_{\theta_{k,\ell}} s_{im} - \nu \partial_{\theta_{k,\ell}} s_{ik} \right) \right]$$
$$+ \frac{1}{2n} \sum_{i=1}^n \frac{1}{|P_y(i)|} \sum_{k \in P_y(i)} \phi'\left( \epsilon_{ki}\psi((1-\nu)s_{ki}) + \sum_{m \notin P_y(i)} \epsilon_{mi}\psi(s_{mi} - \nu s_{ki}) \right)$$
$$\cdot \left[ \epsilon_{ki}\psi'((1-\nu)s_{ki})(1-\nu)\partial_{\theta_{k,\ell}} s_{ki} + \sum_{m \notin P_y(i)} \epsilon_{mi}\psi'(s_{mi} - \nu s_{ki})\left( \partial_{\theta_{k,\ell}} s_{mi} - \nu \partial_{\theta_{k,\ell}} s_{ki} \right) \right] + \partial_{\theta_{k,\ell}} R \tag{42}$$

$$= \frac{1}{2n} \sum_{i=1}^n \frac{1}{|P_x(i)|} \left( \sum_{k \in P_x(i)} \gamma_{ik}\partial_{\theta_{k,\ell}} s_{ik} + \sum_{m \notin P_x(i)} \gamma_{im}\partial_{\theta_{k,\ell}} s_{im} \right)$$
$$+ \frac{1}{2n} \sum_{i=1}^n \frac{1}{|P_y(i)|} \left( \sum_{k \in P_y(i)} \bar{\gamma}_{ik}\partial_{\theta_{k,\ell}} s_{ki} + \sum_{m \notin P_y(i)} \bar{\gamma}_{im}\partial_{\theta_{k,\ell}} s_{mi} \right) + \partial_{\theta_{k,\ell}} R \tag{43}$$

$$= \frac{1}{2n} \sum_{i=1}^n \sum_{j=1}^n \left( \frac{\gamma_{ij}}{|P_x(i)|} + \frac{\bar{\gamma}_{ji}}{|P_y(j)|} \right) \partial_{\theta_{k,\ell}} s_{ij} \tag{44}$$

Define

$$\gamma_{ij} = \begin{cases} \phi'_{ij} \cdot \left( \epsilon_{ij}(1-\nu)\psi'((1-\nu)s_{ij}) - \nu \sum_{m \notin P_x(i)} \epsilon_{im}\psi'(s_{im} - \nu s_{ij}) \right), & \text{if } j \in P_x(i) \\ \sum_{k \in P_x(i)} \phi'_{ik} \cdot (\epsilon_{ij}\psi'(s_{ij} - \nu s_{ik})), & \text{if } j \notin P_x(i) \end{cases} \tag{45}$$

$$\bar{\gamma}_{ij} = \begin{cases} \bar{\phi}'_{ij} \cdot \left( \epsilon_{ji}(1-\nu)\psi'((1-\nu)s_{ji}) - \nu \sum_{m \notin P_y(i)} \epsilon_{mi}\psi'(s_{mi} - \nu s_{ji}) \right), & \text{if } j \in P_y(i) \\ \sum_{k \in P_y(i)} \bar{\phi}'_{ik} \cdot (\epsilon_{ji}\psi'(s_{ji} - \nu s_{ki})), & \text{if } j \notin P_y(i) \end{cases} \tag{46}$$

where we define

$$\phi'_{ij} = \phi' \left( \epsilon_{ij}\psi((1-\nu)s_{ij}) + \sum_{m \notin P_x(i)} \epsilon_{im}\psi(s_{im} - \nu s_{ij}) \right), \tag{47}$$

$$\bar{\phi}'_{ij} = \phi' \left( \epsilon_{ji}\psi(1-\nu)s_{ji} + \sum_{m \notin P_y(i)} \epsilon_{mi}\psi(s_{mi} - \nu s_{ji}) \right) \tag{48}$$

To simplify the notation, we assume that the encoded representations $\mathbf{f}_{\theta_1}(x_i)$ and $\mathbf{f}_{\theta_2}(y_j)$ are already $\ell_2$-normalized. Then the gradient follows that

$$\partial_{\theta_k,\ell}\mathcal{L} = \frac{1}{2n} \sum_{i=1}^{n} \sum_{j=1}^{n} \left( \frac{\gamma_{ij}}{|P_x(i)|} + \frac{\bar{\gamma}_{ji}}{|P_y(j)|} \right) \partial_{\theta_k,\ell}(\mathbf{f}_{\theta_1}^\top(x_i)\mathbf{f}_{\theta_2}(y_j)) + \partial_{\theta_k,\ell}R \tag{49}$$

$$= \partial_{\theta_k,\ell} \left( \sum_{i,j} -S(\gamma)_{ij}(\mathcal{F}_{\theta_1}^\top(\mathbf{X})\mathcal{F}_{\theta_2}(\mathbf{Y}))_{ij} \right) + \partial_{\theta_k,\ell}R \tag{50}$$

where $(\mathcal{F}_{\theta_1}^\top(\mathcal{X})\mathcal{F}_{\theta_2}(\mathcal{Y}))_{ij} = \mathbf{f}_{\theta_1}^\top(x_i)\mathbf{f}_{\theta_2}(y_j)$ denotes the similarity between sample $x_i$ and $y_j$, and $S(\gamma)_{ij}$ denotes the entry in the $i$-th row and $j$-th column of the matrix $S(\gamma)$, which is defined in Equation equation 35.

Note that

$$\text{tr}(A^\top B) = \sum_{i=j} \sum_k (A^\top)_{ik} \cdot B_{kj} = \sum_{ik} A_{ki}B_{ki} \tag{51}$$

Therefore we have

$$-\frac{\partial \mathcal{L}}{\partial \theta_k} = \frac{\partial \, \text{tr}(S(\gamma)^\top \mathcal{F}_{\theta_1}^\top(\mathbf{X})\mathcal{F}_{\theta_2}(\mathbf{Y}))}{\partial \theta_k} - \frac{\partial R(\theta_1,\theta_2)}{\partial \theta_k} \tag{52}$$

$$= \frac{\partial \, \text{tr}(\mathcal{F}_{\theta_1}(\mathbf{X})S(\gamma)\mathcal{F}_{\theta_2}^\top(\mathbf{Y}))}{\partial \theta_k} - \frac{\partial R(\theta_1,\theta_2)}{\partial \theta_k} \tag{53}$$

where $S(\gamma)$ is defined as Equation equation 35.

$\square$

## B.1 Linear Representation Setting

In this setting, the hyper-spherical similarity between a pair $(\mathbf{x}_i, \mathbf{y}_j)$ is computed as the inner product of their $\ell_2$-normalized embeddings:

$$s_{ij} = \langle F_1 \mathbf{x}_i, F_2 \mathbf{y}_j \rangle_{\mathbb{S}^{r-1} \subset \mathbb{R}^r} = \left\langle \frac{F_1 \mathbf{x}_i}{\|F_1 \mathbf{x}_i\|_2}, \frac{F_2 \mathbf{y}_j}{\|F_2 \mathbf{y}_j\|_2} \right\rangle_{\mathbb{R}^r} = \frac{\mathbf{x}_i^\top F_1^\top F_2 \mathbf{y}_j}{\|F_1 \mathbf{x}_i\|_2 \|F_2 \mathbf{y}_j\|_2}. \tag{54}$$

**Proposition 7** *Under the linear setting, the Lemma 4 is specialized as*

$$\frac{\partial \mathcal{L}}{\partial F_k} = -\frac{\partial \, \text{tr}\left( F_1 \mathbf{X} S(\gamma) \mathbf{Y}^\top F_2^\top \right)}{\partial F_k} \bigg|_{\beta=\beta(F_1,F_2)} + \frac{\partial R(F_1, F_2)}{\partial F_k}, \quad k \in \{1,2\} \tag{55}$$

$$= -\frac{\partial \, \text{tr}\left( F_1 C(\gamma) F_2^\top \right)}{\partial F_k} \bigg|_{\beta=\beta(F_1,F_2)} + \frac{\partial R(F_1, F_2)}{\partial F_k}, \quad k \in \{1,2\} \tag{56}$$

where $C(\gamma) = \mathbf{X}S(\gamma)\mathbf{Y}^\top$.

To solve the optimization problem induced by our reformulated objective, we characterize its maximizer in the linear setting. We arrive at the following theorem, which establishes that the convergence of the contrastive loss can be replaced by a closed-form update.

**Theorem 8** (Spectral Characterization (Nakada et al., 2023)) *Consider minimizing the contrastive loss function $\mathcal{L}(F_1, F_2)$, with $R(F_1, F_2) = \frac{\rho}{2}||F_1^T F_2||_F^2$. Then,*

$$\underset{F_1 \in \mathbb{R}^{r \times d_1}, F_2 \in \mathbb{R}^{r \times d_2}}{\arg\min} \mathcal{L}(F_1, F_2) \tag{57}$$

$$= \underset{F_1 \in \mathbb{R}^{r \times d_1}, F_2 \in \mathbb{R}^{r \times d_2}}{\arg\max} \operatorname{tr}\left(F_1 C(\gamma) F_2^\top\right) - (\rho/2)\left\|F_1^\top F_2\right\|_F^2 \tag{58}$$

$$= \left\{ (F_1, F_2) \in \mathbb{R}^{r \times d_1} \times \mathbb{R}^{r \times d_2} : F_1^\top F_2 = \frac{1}{\rho}\sum_{i=1}^r \sigma_i u_i v_i^\top \right\} \tag{59}$$

*where $\{\sigma_i, u_i, v_i\}_{i=1}^r$ are the top-r singular values and vectors of $C(\gamma)$ according to the Eckart–Young–Mirsky theorem.*

*Proof.* Observe that

$$\operatorname{tr}\left(F_1 C(\gamma) F_2^\top\right) - (\rho/2)\left\|F_1^\top F_2\right\|_F^2 \tag{60}$$

$$= \operatorname{tr}\left(F_1 C(\gamma) F_2^\top\right) - \frac{\rho}{2}\operatorname{tr}\left(F_2^\top F_1 F_1^\top F_2\right) \tag{61}$$

$$= \operatorname{tr}\left(F_1 C(\gamma) F_2^\top\right) - \frac{\rho}{2}\operatorname{tr}\left(F_2^\top F_1 F_1^\top F_2\right) - \frac{1}{2\rho}\operatorname{tr}\left(C(\gamma)^\top C(\gamma)\right) + \frac{1}{2\rho}\operatorname{tr}\left(C(\gamma)^\top C(\gamma)\right) \tag{62}$$

$$= \frac{1}{2\rho}\operatorname{tr}\left(C(\gamma)^T C(\gamma)\right) - \frac{\rho}{2}\operatorname{tr}\left[\left(F_1^\top F_2 - \frac{1}{\rho}C(\gamma)\right)^\top \left(F_1^\top F_2 - \frac{1}{\rho}C(\gamma)\right)\right] \tag{63}$$

$$= \frac{1}{2\rho}\left\|C(\gamma)\right\|_F^2 - \frac{\rho}{2}\left\|F_1^\top F_2 - \frac{1}{\rho}(C(\gamma))\right\|_F^2 \tag{64}$$

The first term is constant for fixed $C(\gamma)$, and the second term is minimized at $F_1^\top F_2 = \frac{1}{\rho}C(\gamma)$. Since $F_1 \in \mathbb{R}^{r \times d_1}, F_2 \in \mathbb{R}^{r \times d_2}$, $F_1^\top F_2$ has rank at most $r$. Thus, the minimization can be achieved at $F_1^\top F_2 = \sum_{i=1}^r \sigma_i u_i v_i^\top$ by Eckart–Young–Mirsky theorem for low rank matrix approximation. Here $\{\sigma_i, u_i, v_i\}$ are the top-$r$ singular values and vectors of $S$. $\qquad\square$

In summary, in linear case, the global minimum is attained by projecting the contrastive covariance $C(\gamma)$ onto its top-r singular components. Thus, gradient descent on any loss in the contrastive family $(\phi, \psi)$ merely tracks the dominant singular subspace of $C(\gamma)$. UniCon performs this update in closed form, replacing thousands of SGD steps with one spectral factorization.

**Relationship with (Nakada et al., 2023)** Furthermore, the formulation of (Nakada et al., 2023) can be seen as a special case of our contrastive similarity weight matrix $S(\gamma)$, which arises under the specific assumptions of one-to-one alignment in a linear representation setting and with further restrictions on the functions $\psi$ and $\phi$. Similar to the paper (Nakada et al., 2023), we define

$$\alpha_{ij} \triangleq \epsilon_{ij}\phi'\left(\sum_{m \in [n]} \epsilon_{im}\psi\left(s_{im} - \nu s_{ii}\right)\right)\psi'\left(s_{ij} - \nu s_{ii}\right), \tag{65}$$

$$\bar{\alpha}_{ij} \triangleq \epsilon_{ij}\phi'\left(\sum_{m \in [n]} \epsilon_{im}\psi\left(s_{mi} - \nu s_{ii}\right)\right)\psi'\left(s_{ji} - \nu s_{ii}\right) \tag{66}$$

Then we can derive that when $|P_x(i)| = |P_y(j)| = 1$, for positive pairs $(x_i, y_j)$ with $i = j$,

$$
\begin{aligned}
(\gamma_{ii} + \bar{\gamma}_{ii}) =&\phi'\left(\epsilon_{ii}\psi((1-\nu)s_{ii}) + \sum_{m\neq i}\epsilon_{im}\psi(s_{im} - \nu s_{ii})\right) \\
&\times \left(\epsilon_{ii}(1-\nu)\psi'((1-\nu)s_{ii}) - \nu\sum_{m\neq i}\epsilon_{im}\psi'(s_{im} - \nu s_{ii})\right) \\
&+ \phi'\left(\epsilon_{ii}\psi((1-\nu)s_{ii}) + \sum_{m\neq i}\epsilon_{mi}\psi(s_{mi} - \nu s_{ii})\right) \\
&\times \left(\epsilon_{ii}(1-\nu)\psi'((1-\nu)s_{ii}) - \nu\sum_{m\neq i}\epsilon_{mi}\psi'(s_{mi} - \nu s_{ii})\right) \quad (67) \\
=&\phi'\left(\sum_{m=1}^{n}\epsilon_{im}\psi(s_{im} - \nu s_{ii})\right)\left(\epsilon_{ii}\psi'(s_{ii} - \nu s_{ii}) - \nu\sum_{m=1}^{n}\epsilon_{im}\psi'(s_{im} - \nu s_{ii})\right) \\
&+ \phi'\left(\sum_{m=1}^{n}\epsilon_{mi}\psi(s_{mi} - \nu s_{ii})\right)\left(\epsilon_{ii}\psi'(s_{ii} - \nu s_{ii}) - \nu\sum_{m=1}^{n}\epsilon_{mi}\psi'(s_{mi} - \nu s_{ii})\right) \\
&\qquad\qquad (68) \\
=&\alpha_{ii} + \bar{\alpha}_{ii} - \nu\sum_{m=1}^{n}(\alpha_{im} + \bar{\alpha}_{im}) \quad (69)
\end{aligned}
$$

For negative pairs $(x_i, y_j)$ with $i \neq j$,

$$
\begin{aligned}
\gamma_{ij} + \bar{\gamma}_{ji} =&\phi'\left(\epsilon_{ii}\psi(1-\nu)s_{ii} + \sum_{m\neq i}\epsilon_{im}\psi(s_{im} - \nu s_{ii})\right)(\epsilon_{ij}\psi'(s_{ij} - \nu s_{ii})) \\
&+ \phi'\left(\epsilon_{jj}\psi(1-\nu)s_{jj} + \sum_{m\neq j}\epsilon_{mj}\psi(s_{mj} - \nu s_{jj})\right)(\epsilon_{ij}\psi'(s_{ij} - \nu s_{jj})) \quad (70) \\
=&\epsilon_{ij}\phi'\left(\sum_{m=1}^{n}\epsilon_{im}\psi(s_{im} - \nu s_{ii})\right)\psi'(s_{ij} - \nu s_{ii}) \\
&+ \epsilon_{ij}\phi'\left(\sum_{m=1}^{n}\epsilon_{mj}\psi(s_{mj} - \nu s_{jj})\right)\psi'(s_{ij} - \nu s_{jj}) \quad (71) \\
=&\alpha_{ij} + \bar{\alpha}_{ji} \quad (72)
\end{aligned}
$$

Then we can define

$$
\beta_{ij} = \frac{\alpha_{ij} + \bar{\alpha}_{ji}}{2}, \qquad \beta_i = \nu\sum_{j=1}^{n}\frac{\alpha_{ij} + \bar{\alpha}_{ij}}{2} - \frac{\alpha_{ii} + \bar{\alpha}_{ii}}{2}. \quad (73)
$$

Thus we have

$$
C(\gamma) = XS(\gamma)Y^{\top} = -\frac{1}{2n}\sum_{i=1}^{n}\sum_{j=1}^{n}(\gamma_{ij} + \bar{\gamma}_{ji})x_i y_j^{\top} = \frac{1}{n}\sum_{i=1}^{n}\beta_i x_i y_i^{\top} - \frac{1}{n}\sum_{i\neq j}\beta_{ij}x_i y_j^{\top} \quad (74)
$$

In Nakada et al. (2023), they define the contrastive cross-covariance $S(\beta)$ as:

$$
S(\beta) = \frac{1}{C_n}\sum_{i=1}^{n}\beta_i x_i y_i^{\top} - \frac{1}{C_n}\sum_{i\neq j}\beta_{ij}x_i y_j^{\top}, \quad (75)
$$

$$\beta_{ij} = \frac{\alpha_{ij} + \bar{\alpha}_{ji}}{2}, \qquad \beta_i = \nu \sum_{j \in [n]} \frac{\alpha_{ij} + \bar{\alpha}_{ij}}{2} - 1. \tag{76}$$

Therefore, with $C_n = n$, our $C(\gamma)$ is equivalent to $S(\beta)$ in Nakada et al. (2023), where their $\beta_i$ corresponds to the special case of our definition with identity functions for $\phi$ and $\psi$.

**Understanding $C(\gamma)$:** Consider the trace objective function

$$\mathrm{tr}\big(F_1\, C(\gamma)\, F_2^\top\big) \;=\; \frac{1}{n} \sum_{i,j=1}^{n} -\frac{\gamma_{ij} + \bar{\gamma}_{ji}}{2} \left\langle F_1(\mathbf{x}_i),\, F_2(\mathbf{y}_j) \right\rangle. \tag{77}$$

Every similarity inside a batch is multiplied by a scalar $-\frac{\gamma_{ij} + \bar{\gamma}_{ji}}{2}$:

- $-\frac{\gamma_{ii} + \bar{\gamma}_{ii}}{2}$ on the diagonal strengthens the *attractive* force for the positive pair $(\mathbf{x}_i, \mathbf{y}_i)$;
- $-\frac{\gamma_{ij} + \bar{\gamma}_{ji}}{2}$ $(i \neq j)$ on the off-diagonals weights the *repulsive* force for negative pairs.

where:

$$-\frac{\gamma_{ij} + \bar{\gamma}_{ji}}{2} \;=\; -\frac{\alpha_{ij} + \bar{\alpha}_{ji}}{2}, \qquad -\frac{\gamma_{ii} + \bar{\gamma}_{ii}}{2} \;=\; \nu \sum_{j=1}^{n} \frac{\alpha_{ij} + \bar{\alpha}_{ij}}{2} - \frac{\alpha_{ii} + \bar{\alpha}_{ii}}{2}. \tag{78}$$

- $\alpha_{ij}$ and $\bar{\alpha}_{ij}$ encode the bidirectional importance of the pair $(i,j)$.
- $\nu$ adjusts the influence of positive pairs relative to negatives.

The plus/minus pattern makes the "pull" vs. "push" behaviour of contrastive learning transparent.

## B.2 KERNELIZED CONTRASTIVE ALIGNMENT

**RKHS parameterization.** Let $(\mathcal{H}_{\mathcal{X}}, k_{\mathcal{X}})$ and $(\mathcal{H}_{\mathcal{Y}}, k_{\mathcal{Y}})$ be RKHSs with kernels $k_X, k_Y$ and canonical feature maps

$$\phi_X : X \to \mathcal{H}_X, \quad \phi_X(\mathbf{x}) := k_X(\cdot, \mathbf{x}), \qquad \phi_Y : Y \to \mathcal{H}_Y, \quad \phi_Y(\mathbf{y}) := k_Y(\cdot, \mathbf{y}). \tag{79}$$

They satisfy the reproducing property:

$$\begin{aligned} \forall f \in \mathcal{H}_X,\ \forall \mathbf{x} \in X: \quad & f(\mathbf{x}) = \langle f, \phi_X(\mathbf{x}) \rangle_{\mathcal{H}_X}, \\ \forall g \in \mathcal{H}_Y,\ \forall \mathbf{y} \in Y: \quad & g(\mathbf{y}) = \langle g, \phi_Y(\mathbf{y}) \rangle_{\mathcal{H}_Y}. \end{aligned} \tag{80}$$

In particular,

$$\langle \phi_X(\mathbf{x}_i), \phi_X(\mathbf{x}) \rangle_{\mathcal{H}_X} = k_X(\mathbf{x}_i, \mathbf{x}), \qquad \langle \phi_Y(\mathbf{y}_j), \phi_Y(\mathbf{y}) \rangle_{\mathcal{H}_Y} = k_Y(\mathbf{y}_j, \mathbf{y}). \tag{81}$$

By the representer theorem, for each output coordinate $a = 1, \ldots, r$,

$$f_{\theta_1}^{(a)}(\cdot) = \sum_{i=1}^{n} A_{ia}\, k_X(\mathbf{x}_i, \cdot), \qquad f_{\theta_2}^{(a)}(\cdot) = \sum_{j=1}^{n} B_{ja}\, k_Y(\mathbf{y}_j, \cdot), \tag{82}$$

and we stack coefficients into matrices $A, B \in \mathbb{R}^{n \times r}$ (*column $a$ stores the coefficients of coordinate $a$*).

Define

$$\kappa_X(\mathbf{x}) := \big[ k_X(\mathbf{x}_1, \mathbf{x}), \ldots, k_X(\mathbf{x}_n, \mathbf{x}) \big]^\top \in \mathbb{R}^n, \qquad \kappa_Y(\mathbf{y}) := \big[ k_Y(\mathbf{y}_1, \mathbf{y}), \ldots, k_Y(\mathbf{y}_n, \mathbf{y}) \big]^\top \in \mathbb{R}^n. \tag{83}$$

Then the $r$-dimensional encoder outputs at a point are

$$\mathbf{f}_{\theta_1}(\mathbf{x}) := \big( f_{\theta_1}^{(1)}(\mathbf{x}), \ldots, f_{\theta_1}^{(r)}(\mathbf{x}) \big)^\top = A^\top \kappa_X(\mathbf{x}) \in \mathbb{R}^r, \qquad \mathbf{f}_{\theta_2}(\mathbf{y}) = B^\top \kappa_Y(\mathbf{y}) \in \mathbb{R}^r. \tag{84}$$

- Theoretically, in infinite-dimensional RKHS,

$$f_{\theta_1}^{(a)}(\mathbf{x}) = \left\langle \sum_{i=1}^{n} A_{ia}\, k_X(\mathbf{x}_i, \cdot),\, \phi_X(\mathbf{x}) \right\rangle_{\mathcal{H}_X}. \tag{85}$$

- Computationally, in finite $n$-dimensional representation,

$$f_{\theta_1}^{(a)}(\mathbf{x}) = A_{\cdot a}^{\top}\, \kappa_X(\mathbf{x}). \tag{86}$$

Let the Gram matrices be

$$K_X = [k_X(x_i, x_j)]_{i,j=1}^{n} \in \mathbb{R}^{n \times n}, \qquad K_Y = [k_Y(y_i, y_j)]_{i,j=1}^{n} \in \mathbb{R}^{n \times n}. \tag{87}$$

Stacking the $n$ samples as columns, the $r \times n$ batch embeddings are

$$\mathcal{F}_{\theta_1}(X) = A^{\top} K_X \in \mathbb{R}^{r \times n}, \qquad \mathcal{F}_{\theta_2}(Y) = B^{\top} K_Y \in \mathbb{R}^{r \times n}. \tag{88}$$

**Inner products and induced norms.** For any $c, d \in \mathbb{R}^n$,

$$\left\langle \sum_i c_i\, \phi_X(\mathbf{x}_i),\, \sum_j d_j\, \phi_X(\mathbf{x}_j) \right\rangle_{\mathcal{H}_X} = c^{\top} K_X d, \qquad \left\| \sum_i c_i\, \phi_X(\mathbf{x}_i) \right\|_{\mathcal{H}_X}^{2} = c^{\top} K_X c, \tag{89}$$

and similarly with $K_Y$.

Therefore, the total RKHS norm of the $r$ output coordinates is

$$\sum_{a=1}^{r} \| f_{\theta_1}^{(a)} \|_{\mathcal{H}_X}^{2} = \sum_{a=1}^{r} A_{\cdot a}^{\top} K_X A_{\cdot a} = \operatorname{tr}(A^{\top} K_X A), \tag{90}$$

and analogously for $B$ with $K_Y$.

**Similarity.** For two samples $(\mathbf{x}_i, \mathbf{y}_j)$ the predicted similarity is

$$\langle \mathbf{f}_{\theta_1}(\mathbf{x}_i), \mathbf{f}_{\theta_2}(\mathbf{y}_j) \rangle_{\mathbb{R}^r} = \kappa_X(\mathbf{x}_i)^{\top} A B^{\top} \kappa_Y(\mathbf{y}_j), \tag{91}$$

which is exactly the $(i, j)$ entry of

$$S_{\text{pred}} := \mathcal{F}_{\theta_1}(X)^{\top} \mathcal{F}_{\theta_2}(Y) = K_X A B^{\top} K_Y \in \mathbb{R}^{n \times n}. \tag{92}$$

with entry $[S_{\text{pred}}]_{ij} = \langle \mathcal{F}_{\theta_1}(x_i), \mathcal{F}_{\theta_2}(y_j) \rangle_{\mathbb{R}^r}$.

**Definition 11** (Kernel cross–covariance regularizer). *Define*

$$\mathcal{R}_{\times}(A, B) := \operatorname{tr}\big(A^{\top} K_X A\, B^{\top} K_Y B\big) = \big\| K_X^{1/2} A B^{\top} K_Y^{1/2} \big\|_F^{2}. \tag{93}$$

The second equality in equation 93 follows from the identity $\|A'^{\top} B'\|_F^2 = \operatorname{tr}(A'^{\top} A' B'^{\top} B')$ with $A' = A K_X^{1/2}$ and $B' = B K_Y^{1/2}$.

In the linear case, $\mathbf{f}_{\theta_1}(\mathbf{X}) = F_1 \mathbf{X}$. Let $w^{(a)\top}$ be the $a$-th row vector of $F_1$, then $f_{\theta_1}^{(a)}(\mathbf{x}) = w^{(a)\top}\mathbf{x}$, thus

$$\begin{aligned}
\sum_{a=1}^{r} \left\| f_{\theta_1}^{(a)} \right\|_{\mathcal{H}_X}^{2} &= \sum_{a=1}^{r} \langle f_{\theta_1}^{(a)}, f_{\theta_1}^{(a)} \rangle_{\mathcal{H}_X} \\
&= \sum_{a=1}^{r} \langle w^{(a)}, w^{(a)} \rangle \\
&= \sum_{a=1}^{r} \| w^{(a)} \|_2^2 \\
&= \| F_1 \|_F^2
\end{aligned} \tag{94}$$

**Proposition 12** (Linear-kernel reduction). *Suppose* $k_X(x, x') = \langle x, x' \rangle$ *and* $k_Y(y, y') = \langle y, y' \rangle$, *and let the sample matrices be* $X = [x_1, \ldots, x_n] \in \mathbb{R}^{d_1 \times n}$, $Y = [y_1, \ldots, y_n] \in \mathbb{R}^{d_2 \times n}$. *Then* $K_X = X^\top X$ *and* $K_Y = Y^\top Y$, *and with*

$$F_1 := A^\top X^\top \in \mathbb{R}^{r \times d_1}, \qquad F_2 := B^\top Y^\top \in \mathbb{R}^{r \times d_2}, \tag{95}$$

*we have*

$$\mathcal{R}_\times(A, B) = \mathrm{tr}(A^\top K_X A \; B^\top K_Y B) = \mathrm{tr}(F_1 F_1^\top F_2 F_2^\top) = \|F_1^\top F_2\|_F^2. \tag{96}$$

*Proof.* With $K_X = X^\top X$ and $K_Y = Y^\top Y$ we compute

$$F_1 F_1^\top = (A^\top X^\top)(XA) = A^\top(X^\top X)A = A^\top K_X A,$$
$$F_2 F_2^\top = (B^\top Y^\top)(YB) = B^\top(Y^\top Y)B = B^\top K_Y B.$$

Therefore,

$$\mathrm{tr}(F_1 F_1^\top F_2 F_2^\top) = \mathrm{tr}(A^\top K_X A \; B^\top K_Y B) = \mathcal{R}_\times(A, B).$$

$\square$

**Theorem 13** (Kernelized spetral charaterization (unified form)). *Let* $\rho > 0$ *and define the regularizer* $R(A, B) = \|(K_X^{1/2}A)^\top(K_Y^{1/2}B)\|_F^2$. *Then minimizing the contrastive loss is equivalent to the kernelized maximization*

$$\max_{A, B \in \mathbb{R}^{n \times r}} \quad \mathrm{tr}(A^\top K_X S(\gamma) K_Y B) - \frac{\rho}{2}\|(K_X^{1/2}A)^\top(K_Y^{1/2}B)\|_F^2. \tag{97}$$

*Let*

$$A' := A^\top K_X^{1/2} \in \mathbb{R}^{r \times n}, \quad B' := B^\top K_Y^{1/2} \in \mathbb{R}^{r \times n}, \quad M := K_X^{1/2} S(\gamma) K_Y^{1/2} \in \mathbb{R}^{n \times n}. \tag{98}$$

*Then equation 97 rewrites*

$$\max_{A', B'} \; \mathrm{tr}(A' M B'^\top) - \frac{\rho}{2}\|A'^\top B'\|_F^2. \tag{99}$$

*If* $M = U\Sigma V^\top$ *is an SVD and* $M_r = \sum_{i=1}^r \sigma_i u_i v_i^\top$ *its best rank-$r$ approximation by Eckart–Young–Mirsky theorem, then all maximizers satisfy the relation*

$$(A')^\top B' = \frac{1}{\rho} M_r \quad \Longleftrightarrow \quad A B^\top = \frac{1}{\rho} K_X^{-1/2} M_r K_Y^{-1/2}. \tag{100}$$

*(If $K_X$ or $K_Y$ is singular, replace inverse square roots by Moore–Penrose pseudo–inverse square roots.)*

*Proof.* Insert $K_X^{1/2} K_X^{1/2} = K_X$ and $K_Y^{1/2} K_Y^{1/2} = K_Y$ into the trace term and set $A' = A^\top K_X^{1/2}$, $B' = B^\top K_Y^{1/2}$, $M = K_X^{1/2} S(\gamma) K_Y^{1/2}$, to obtain

$$\mathrm{tr}\left(A^\top K_X S(\gamma) K_Y B\right) - (\rho/2)\|(K_X^{1/2}A)^\top(K_Y^{1/2}B)\|_F^2 \tag{101}$$

$$= \mathrm{tr}\left(A^\top K_X^{\frac{1}{2}} K_X^{\frac{1}{2}} S(\gamma) K_Y^{\frac{1}{2}} K_Y^{\frac{1}{2}} B\right) - (\rho/2)\|(A^\top K_X^{1/2})^\top(B^\top K_Y^{1/2})\|_F^2 \tag{102}$$

$$= \mathrm{tr}\left(A' M B'^\top\right) - (\rho/2)\|A'^\top B'\|_F^2 \tag{103}$$

Now same as steps in proof of Theorem8, complete the square in $A'^\top B'$ to see that the maximizer aligns the column spaces of $A', B'$ with the top singular vectors of $M$, yielding $A'^\top B' = \rho^{-1} M_r$. Undo the change of variables to get equation 100.

By Theorem 8, the optimal solution satisfies

$$\left\{(A, B) : A'^\top B' = \frac{1}{\rho} M_r(\gamma)\right\} \tag{104}$$

$$\left\{(A, B) : (A^\top K_X^{\frac{1}{2}})^\top(B^\top K_Y^{\frac{1}{2}}) = \frac{1}{\rho} K_X^{\frac{1}{2}} S(\gamma) K_Y^{\frac{1}{2}}\right\} \tag{105}$$

$$\left\{(A, B) : K_X^{\frac{1}{2}} A B^\top K_Y^{\frac{1}{2}} = \frac{1}{\rho} K_X^{\frac{1}{2}} S(\gamma) K_Y^{\frac{1}{2}}\right\} \tag{106}$$

$$\left\{(A, B) : A B^\top = \frac{1}{\rho} K_X^{-\frac{1}{2}} K_X^{\frac{1}{2}} S(\gamma) K_Y^{\frac{1}{2}} K_Y^{-\frac{1}{2}}\right\} \tag{107}$$

If $K_X$ or $K_Y$ is singular, use Moore–Penrose pseudoinverses $K_X^{+1/2}$, $K_Y^{+1/2}$. $\square$

One explicit optimal choice is $A^\star = K_X^{-1/2} U_r$ and $B^\star = K_Y^{-1/2} V_r \Sigma_r / \rho$, where $U_r = [u_1, \ldots, u_r]$, $V_r = [v_1, \ldots, v_r]$, $\Sigma_r = \text{diag}(\sigma_1, \ldots, \sigma_r)$ are SVD of $M$.

**Corollary 14** (Kernel inference (out-of-sample)). *Let $\mathcal{D} = \{(\mathbf{x}_i, \mathbf{y}_i)\}_{i=1}^n$ be the reference batch used to build contrastive similarity $S(\gamma)$ and let $k$ be an positive-definite kernel. For a new pair $(x^*, y^*)$, set $\kappa_X(x^*) = [k_X(x_1, x^*), \ldots, k_X(x_n, x^*)]^\top$ and $\kappa_Y(y^*) = [k_Y(y_1, y^*), \ldots, k_Y(y_n, y^*)]^\top$. With an optimal $(A^\star, B^\star)$ from Theorem 9,*

$$\mathbf{f}_{\theta_1}(x^*) = (A^\star)^\top \kappa_X(x^*), \qquad \mathbf{f}_{\theta_2}(y^*) = (B^\star)^\top \kappa_Y(y^*), \tag{108}$$

*and the similarity*

$$s(\mathbf{x}^*, \mathbf{y}^*) = \frac{\kappa_X(x^*)^\top A^\star B^{\star\top} \kappa_Y(y^*)}{\|A^{\star\top} \kappa_X(x^*)\|_2 \|B^{\star\top} \kappa_Y(y^*)\|_2} \tag{109}$$

## C  IMPLEMENTATION

The complete code for all experiments will be made publicly available on GitHub https://github.com/suihangke/UniCon.git.

### C.1  COMPUTATION OF $S(\gamma)$

In this section, we provide the PyTorch implementation of the contrastive similarity weight matrix $S(\gamma)$ computation used in UniCon. The function below computes the matrix $S(\gamma)$ in one-to-one paired settings, and a generalized many-to-many settings, where similarity values $s_{ij}$ are used to form the weighting terms $\gamma_{ij}$. We have also provided a complete pseudocode outlining the training pipeline of UniCon in Algorithm 1 2, which offers a step-by-step description of our method's implementation to facilitate reproducibility and deeper understanding.

**Algorithm Discussion.**  In UniCon, our efficiency claim refers to the rapid stabilization of the mapping $((A, B))$ parameters toward the desired alignment subspace through derived-form spectral updates, which bypass many small gradient steps. Specifically, we reformulate the minimization of the contrastive loss as an equivalent maximization problem, using the proposed contrastive similarity matrix $S(\gamma)$. We will state in the unified nonlinear form, as the notations can be reduced to linear case as analysis in Section 3.3. The general objective is $\max tr(A^T K_X S(\gamma) K_Y B) - \frac{\rho}{2} \|(K_X^{1/2} A)^T (K_Y^{1/2} B)\|_F^2$. The optimal solutions satisfy $AB^T = \frac{1}{\rho} K_X^{-1/2} [K_X^{1/2} S(\gamma) K_Y^{1/2}]_r K_Y^{-1/2}$ where $[\cdot]_r$ denotes the best rank-$r$ approximation. Note that $S(\gamma)$ is itself a function of $(A, B)$. Therefore, the overall optimization becomes a fixed-point problem. We solve it via an iterative procedure, such as,

$$A^{(t+1)}, B^{(t+1)} \leftarrow \frac{1}{\rho} K_X^{-1/2} [K_X^{1/2} S(\gamma; A^{(t)}, B^{(t)}) K_Y^{1/2}]_r K_Y^{-1/2}.$$

In this process, we observe rapid convergence of $(A, B)$ to a stable solution. This rapid stabilization, subspace convergence, is what we refer to as efficiency.

```python
def compute_S_gamma(
    s, data1_batch, data2_batch,
    tau=1.0, nu=0.1,
    psi=torch.exp, phi=torch.log1p,
    epsilon_ij=1, epsilon_ii=1,
    diff_psi=torch.exp,
    diff_phi=lambda x, eps=1e-8: 1.0 / (1.0 + x + eps)
):
    n = data1_batch.size(0)

    # Build epsilon mask for weighting
    epsilon = epsilon_ii * torch.eye(n, device=s.device)
    epsilon += epsilon_ij * (1 - torch.eye(n, device=s.device))

    # Row-wise similarity terms: s_ij - nu * s_ii
    s_diag_row = torch.diag(s).unsqueeze(1).expand(-1, n)
    s_nu_row = s - nu * s_diag_row
```

```
18      psi_terms = psi(s_nu_row)
19      sum_psi_terms = torch.sum(epsilon * psi_terms, dim=1, keepdim=True)
20      diff_phi_terms = diff_phi(sum_psi_terms)
21      diff_psi_terms = diff_psi(s_nu_row)
22      alpha = epsilon * diff_phi_terms * diff_psi_terms
23
24      # Column-wise similarity terms: s_ji - nu * s_ii
25      s_diag_col = torch.diag(s).expand(n, n)
26      s_nu_col = s - nu * s_diag_col
27      psi_terms_bar = psi(s_nu_col)
28      sum_psi_terms_bar = torch.sum(epsilon * psi_terms_bar.T, dim=1,
        keepdim=True)
29      diff_phi_terms_bar = diff_phi(sum_psi_terms_bar)
30      diff_psi_terms_bar = diff_psi(s_nu_col.T)
31      alpha_bar = epsilon * diff_phi_terms_bar * diff_psi_terms_bar
32
33      # Compute S_gamma weights
34      S_ij = (alpha + alpha_bar.t()) / 2
35      S_i = nu * torch.sum((alpha + alpha_bar) / 2, dim=1) - torch.diag(
        alpha + alpha_bar) / 2
36
37
38      S_gamma = -S_ij / n
39      S_gamma[range(n), range(n)] = S_i / n
40
41      return S_gamma
```

Listing 1: Contrastive Smilarity Weight Matrix Computation (one-to-one case)

```
1   def compute_S_gamma_generalized(s, pos_mask, normalized_data1_batch,
        normalized_data2_batch, nu=1.5, tau=1.0,psi=torch.exp, phi=torch.
        log1p, epsilon_ij=1, epsilon_ii=1, diff_psi=torch.exp, diff_phi=
        lambda x, eps=1e-8: 1.0 / (1.0 + x + eps)):
2       n = s.shape[0]
3       device = s.device
4
5       # Create epsilon matrix
6       epsilon = torch.ones(n, n, device=device) * epsilon_ij
7       epsilon.fill_diagonal_(epsilon_ii)
8
9       # Create masks
10      neg_mask_base = ~pos_mask
11
12      # Expand tensors for broadcasting: (n, n, n) where dim 0 is i, dim 1
        is j, dim 2 is m
13      s_i_m = s.unsqueeze(1).expand(n, n, n)   # s[i,m]
14      s_i_j = s.unsqueeze(2).expand(n, n, n)   # s[i,j]
15      epsilon_i_m = epsilon.unsqueeze(1).expand(n, n, n)   # epsilon[i,m]
16      mask_not_i = neg_mask_base.unsqueeze(1).expand(n, n, n)   # mask for m
        not in [i]
17      # Compute psi terms: epsilon[i,m] * psi(s[i,m] - nu * s[i,j])
18      psi_terms = epsilon_i_m * psi(s_i_m - nu * s_i_j) * mask_not_i
19      sum_psi = psi_terms.sum(dim=2)   # (n, n), sum over m
20      # Compute diff_psi terms
21      diff_psi_terms = epsilon_i_m * diff_psi(s_i_m - nu * s_i_j) *
        mask_not_i
22      sum_diff_psi = diff_psi_terms.sum(dim=2)   # (n, n)
23      # Phi arguments
24      phi_args = epsilon * psi((1 - nu) * s) + sum_psi   # (n, n)
25      # Gamma for positive samples
26      gamma_pos = diff_phi(phi_args) * (
27          epsilon * diff_psi((1 - nu) * s) * (1 - nu) - nu * sum_diff_psi
28      )
29
30      diff_phi_i_k = diff_phi(phi_args).unsqueeze(1).expand(n, n, n)
```

```python
31      gamma_neg_k = diff_phi_i_k * (epsilon * diff_psi(s_i_j - nu * s_i_m))
32      pos_mask_k = pos_mask.unsqueeze(1).expand(n, n, n)
33      gamma_neg = (gamma_neg_k * pos_mask_k).sum(dim=2)
34
35      # For gamma_bar positive samples
36      s_m_i = s.T.unsqueeze(1).expand(n, n, n)  # s[m,i]
37      s_j_i = s.T.unsqueeze(2).expand(n, n, n)  # s[j,i]
38      epsilon_m_i = epsilon.T.unsqueeze(1).expand(n, n, n)  # epsilon[m,i]
39      # Compute psi terms for gamma_bar
40      psi_terms_bar = epsilon_m_i * psi(s_m_i - nu * s_j_i) * mask_not_i
41      sum_psi_bar = psi_terms_bar.sum(dim=2)  # (n, n), sum over m
42      # Compute diff_psi terms for gamma_bar (no epsilon in second sum)
43      diff_psi_terms_bar = diff_psi(s_m_i - nu * s_j_i) * mask_not_i
44      sum_diff_psi_bar = diff_psi_terms_bar.sum(dim=2)  # (n, n)
45      # Phi arguments for gamma_bar
46      phi_args_bar = epsilon.T * psi((1 - nu) * s.T) + sum_psi_bar  # (n, n
        )
47      # gamma_bar for positive samples
48      gamma_bar_pos = diff_phi(phi_args_bar) * (
49          epsilon.T * diff_psi((1 - nu) * s.T) * (1 - nu) - nu *
        sum_diff_psi_bar
50      )
51
52      diff_phi_i_k = diff_phi(phi_args_bar).unsqueeze(1).expand(n, n, n)
53      gamma_bar_neg_k = diff_phi_i_k * (epsilon.T * diff_psi(s_j_i - nu *
        s_m_i))
54      pos_mask_k = pos_mask.unsqueeze(1).expand(n, n, n)
55      gamma_bar_neg = (gamma_bar_neg_k * pos_mask_k).sum(dim=2)
56
57      # ========== Combine positive and negative samples ==========
58      gamma = torch.where(pos_mask, gamma_pos, gamma_neg)
59      gamma_bar = torch.where(pos_mask, gamma_bar_pos, gamma_bar_neg)
60
61      pos_mask_row_sum = pos_mask.sum(dim=1)
62      pos_mask_col_sum = pos_mask.sum(dim=0)
63      pos_mask_row_sum_expanded = pos_mask_row_sum.unsqueeze(1).expand(n, n
        )
64      pos_mask_col_sum_expanded = pos_mask_col_sum.unsqueeze(0).expand(n, n
        )
65
66      S_gamma = -(gamma / pos_mask_row_sum_expanded + gamma_bar.T /
        pos_mask_col_sum_expanded) / 2
67
68      C_n = n  # Normalization constant
69      S_gamma = S_gamma / C_n
70
71      return S_gamma
```

Listing 2: Contrastive Similarity Weight Matrix Computation (generalized, support many-to-many matching)

## C.2 EXPERIMENT DETAILS AND CONVERGENCE VISUALIZATIONS

We present additional plots to illustrate the convergence behavior of the SGD-CLIP baseline across three experimental settings: synthetic latent factor models, unimodal image clustering, and multimodal image–text retrieval.

These visualizations confirm that our reported SGD-CLIP performance is after sufficient training, providing a fair comparison against our proposed Unicon method. While SGD-CLIP ultimately achieves high accuracy, it requires many iterations to converge—underscoring the computational inefficiency of iterative optimization when compared to the efficient closed form update of Unicon.

---

**Algorithm 1** Training Pipeline for Linear Case

---

**Require:** Initial $F_1^0, F_2^0$, dataset $\{(X_i, Y_i)\}_{i=1}^N$
**Ensure:** Trained $F_1$ and $F_2$
 1: Initialize $F_1 \leftarrow F_1^0$, $F_2 \leftarrow F_2^0$
 2: **while** not converged **do**
 3:   $(F_1^\top F_2)\_\text{sum} \leftarrow 0$ {accumulator for batch-wise $F_1^\top F_2$}
 4:   **for** each batch $(X_i, Y_i)$ **do**
 5:     similarity $\leftarrow (F_1 X_i)^\top (F_2 Y_i)$
 6:     $S(\gamma) \leftarrow \text{compute\_S\_gamma}(\text{similarity}, X_i, Y_i)$
 7:     $(F_1^\top F_2)_i \leftarrow \frac{1}{\rho} X_i S(\gamma) Y_i^\top$
 8:     $\text{weight}_i \leftarrow \text{validation}((F_1^\top F_2)_i)$
 9:     $(F_1^\top F_2)\_\text{sum} \leftarrow (F_1^\top F_2)\_\text{sum} + (F_1^\top F_2)_i^\top \cdot \text{weight}_i$
10:   **end for**
11:   Update $F_1, F_2$ based on aggregated $(F_1^\top F_2)\_\text{sum}$
12:   **decompose** $F_1$ **and** $F_2$
13: **end while**=0

---

---

**Algorithm 2** Training Pipeline for Nonlinear Case

---

**Require:** Initial $A_0, B_0$, dataset $\{(X_i, Y_i)\}_{i=1}^N$
**Ensure:** Trained $A$ and $B$
 1: Initialize $A \leftarrow A_0$, $B \leftarrow B_0$
 2: **while** not converged **do**
 3:   $AB\_\text{sum} \leftarrow 0$ {accumulator for batch-wise $AB^\top$}
 4:   **for** each batch $(X_i, Y_i)$ **do**
 5:     $K_{X_i}, K_{Y_i} \leftarrow$ kernel matrices of $X_i, Y_i$
 6:     similarity $\leftarrow (A^\top K_{X_i})^\top (B^\top K_{Y_i})$
 7:     $S(\gamma) \leftarrow \text{compute\_S\_gamma}(\text{similarity}, X_i, Y_i)$
 8:     $(AB^\top)_i \leftarrow \frac{1}{\rho} K_{X_i}^{-1/2} \left[ K_{X_i}^{1/2} S(\gamma; A^{(t)}, B^{(t)}) K_{Y_i}^{1/2} \right]_r K_{Y_i}^{-1/2}$
 9:     $\text{weight}_i \leftarrow \text{validation}((AB^\top)_i)$
10:     $AB\_\text{sum} \leftarrow AB\_\text{sum} + (AB^\top)_i \cdot \text{weight}_i$
11:   **end for**
12:   Update $A, B$ based on aggregated $AB\_\text{sum}$
13:   **decompose** $A$ **and** $B$
14: **end while**=0

---

**Synthetic Setting: Linear Latent-Factor Model.** We generate synthetic data from latent vectors $\mathbf{z} \in \mathbb{R}^r$ that are sampled around $K = 3$ cluster centers in latent space.

The observed pairs $(\mathbf{x}, \mathbf{y})$ are linearly projected from $\mathbf{z}$ using orthogonal matrices, followed by additive Gaussian noise:

$$\mathbf{x} = U_1 \mathbf{z} + \xi_1, \qquad \mathbf{y} = U_2 \mathbf{z} + \xi_2. \tag{110}$$

Here, $U_1 \in \mathbb{R}^{d_1 \times r}$ and $U_2 \in \mathbb{R}^{d_2 \times r}$ are orthogonal projections sampled from the Haar measure on $\mathbb{O}_{d_1}$ and $\mathbb{O}_{d_2}$ respectively, with $d_1 = 40, d_2 = 30, r = 10$. Noise vectors $\xi_1, \xi_2$ are sampled from $\mathcal{N}(0, SNR^2)$, with $SNR = 0.3$. Each of the $N = 600$ training samples is drawn from one of $K = 3$ clusters in latent space. Figure C1 shows the accuracy score of SGD-CLIP method converges across training epochs.

**Synthetic Setting: Nonlinear Latent-Factor Model.** We further evaluate the method under a nonlinear transformation of the latent space:

$$x = \tanh(U_1 z + \xi_1), \qquad y = \tanh(U_2 z + \xi_2), \tag{111}$$

where $U_1, U_2$ are again uniformly sampled from orthogonal groups $\mathbb{O}_{d_1}$ and $\mathbb{O}_{d_2}$, taking the first $r$ columns. The additive Gaussian noise is drawn from $\mathcal{N}(0, \text{SNR}^2)$ with SNR $= 0.3$.

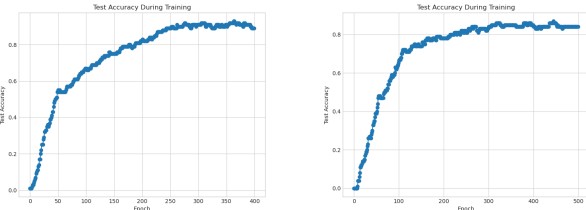

(a) Linear latent factor model    (b) Nonlinear latent factor model

Figure C1: Convergence of SGD-CLIP. Training accuracy over epochs for linear and nonlinear synthetic settings.

**Unimodal Classification (CIFAR-10).** Figure C2 reports the training loss of SGD-CLIP in the CIFAR-10 image clustering task.

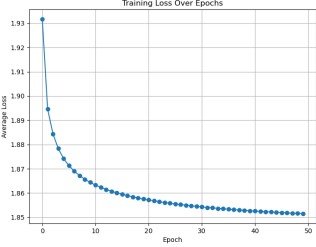

Figure C2: Training loss of SGD-CLIP on CIFAR-10. Unimodal image clustering task with frozen ResNet-18 features.

**Multimodal Retrieval.** On Flickr30K, we shuffle the dataset into to 25426 (80%), 3178 (10%), and 3179(10%), for train/validation/test. Every image $x$ and text $y$ is embedded once with a) **ResNet-18** (He et al., 2016) for images + **Sentence-BERT** (`all-mpnet-base-v2`) (Reimers & Gurevych, 2019); b) **ResNet-50** + Sentence-BERT; c) the **CLIP ViT-B/32** model for visual-textual feature extraction as frozen backbone. Matching is performed in a shared embedding space of dimension $r = 128$ with $\tau = 1$. SGD-CLIP runs for 50 epochs, and run-time is measured wall-clock. Figure C3 shows loss curves for SGD-CLIP trained on Flickr30K across three backbones: ResNet-18, ResNet-50, and CLIP ViT-B/32. Despite faster convergence with stronger backbones, all variants require many epochs to reach stable loss values, whereas UniCon completes alignment more efficiently.

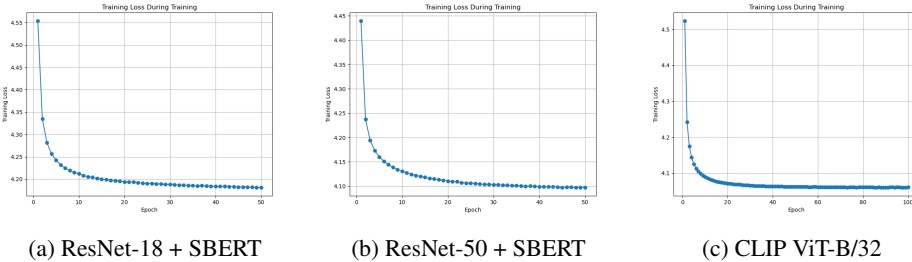

(a) ResNet-18 + SBERT          (b) ResNet-50 + SBERT          (c) CLIP ViT-B/32

Figure C3: Training loss of SGD-CLIP on Flickr30K. Loss curves for three backbone architectures in multimodal alignment.

On **MSCOCO**, we follow the standard retrieval protocol on MSCOCO. The training set contains **82,783** images, each paired with **5** human captions. We validate on **40,504** image–text pairs and report test results on **5,000** held-out pairs. We report Recall@1 and Recall@10 for both directions.

To further ensure sufficiency of training, we re-evaluated validation accuracies on MSCOCO with backbone Resnet 50 + SBERT in FigureC4: it even begin to decline after the peak, suggesting potential overfitting with additional training. Specifically, on MSCOCO, we trained SGD for 1000 epochs and observed that the best validation performance is already reached around epoch 300. Beyond this point, the model does not improve further. For comparison, we extended UniCon to 20 iterations across all batches. We observed that model norms stabilize after just 2 iterations, with only minimal fluctuations thereafter.

Table 3: **Image-text retrieval on MSCOCO**. We report Recall@1 and Recall@10 for both image→text and text→image directions. **UniCon** achieves superior accuracy to SGD–CLIP with ∼96–461× faster training.

| Backbone | Method | Train time | Image→Text | | Text→Image | | Average | |
|---|---|---|---|---|---|---|---|---|
| | | | R@1 | R@10 | R@1 | R@10 | R@1 | R@10 |
| RN-50 + SBERT | SGD–CLIP | 5121.72 s | .053 | .253 | .060 | .286 | .057 | .270 |
| | **UniCon** | **11.11 s** | **.105** | **.388** | **.129** | **.439** | **.117** | **.414** |
| CLIP ViT-B/32 | SGD–CLIP | 1066.60 s | .128 | .415 | .123 | .427 | .126 | .421 |
| | **UniCon** | **11.15 s** | **.329** | **.685** | **.292** | **.644** | **.311** | **.665** |

Table 4: **Zero-shot image–text retrieval on FLICKR30K** (trained on MSCOCO, no fine-tuning). We report Recall@5 and Recall@10 for both directions; higher is better.

| Backbone | R@5 | | | R@10 | | |
|---|---|---|---|---|---|---|
| | I→T | T→I | Avg | I→T | T→I | Avg |
| RN-50 + SBERT | .171 | .249 | .210 | .261 | .353 | .307 |
| **CLIP ViT-B/32** | **.808** | **.766** | **.787** | **.879** | **.848** | **.863** |

### C.3  SENSITIVITY ANALYSIS

**Robustness to batch size.**  We discuss that the batch size $n$ does not significantly affect the total model performance. We extensively evaluated the effect of batch sizes on all the experiment settings, showing robustness of UniCon to batch size variations. For multimodal alignment, we experimented retrieval task using Flickr30k and MSCOCO varying the batch size across [100, 500, 1000, 10000, 20000]. For unimodal alignment, we experimented clustering using CIFAR-10 with batch size [200,300,400]. We observed that performance metrics remained nearly identical across

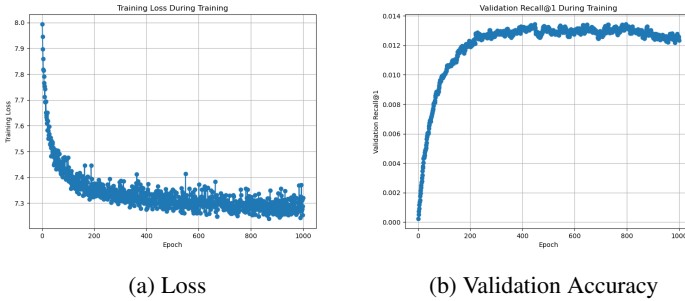

(a) Loss            (b) Validation Accuracy

Figure C4: Training loss and validation accuracy of SGD-CLIP on MSCOCO trained for 1000 epochs to check convergence.

these ranges. Importantly, the retrieval performance was robust to batch size variations, which implies data efficiency.

**Batch aggregation strategy** To reduce memory and computation overhead, we adopt a batch-wise training strategy. We evaluate several strategies to aggregate multiple batch-level models into a global predictor, including:

- Accuracy-weighted fusion: Normalize validation accuracies $a_i$ to weights $w_i = \frac{a_i}{\sum_i a_i}$ and linearly combine predictions.
- Softmax-accuracy weighted fusion: Apply a softmax over $\{a_i\}$ to smooth weights.
- Majority voting: Select the most frequent prediction across batch models.

From our experiments on both unimodal and multimodal settings, we observe that different aggregation strategies yield similar performance, with variations within a 1–2% gap. Its performance under extremely biased or imbalanced data distributions remains an open question.

We also evaluate the impact of statistical differences between training batches on CIFAR-10. For each batch, we form paired inputs data1 and data2 under two settings: (1) **Random**: For each sample pair, we independently sample classes for data1 and data2, then select an image from each chosen class. This independent sampling creates varying class distributions across batches and introduces inter-batch differences. (2) **Balanced**: We iterate through all 10 classes within per batch, sampling two images per class for data1 and data2, ensuring balanced and identical class distributions within and across batches. From our experiments on unimodal settings on CIFAR10, we find that the random and balanced sampling strategies yield similar performance, with differences within 1–2%.

## C.4 NUMERICAL STABILIZATION FOR SPECTRAL UPDATES IN UNICON

**Stabilized SVD.** We analyze the numerical profile of $C(\gamma) \in \mathbb{R}^{d_1 \times d_2}$. On large, high–dimensional tasks, the raw $C(\gamma)$ often exhibits rapid spectral decay, small singular–value gaps, and large effective condition numbers. To stabilize the closed–form spectral step, we tested the following techniques:

- **Tikhonov regularization.** Add $\lambda I_{d_1}$ to $C_{(\gamma)}$ to improve conditioning and stabilize SVD.
- **Randomized SVD with power iterations** (Halko et al., 2011). Use randomized SVD with power iterations to efficiently extract the top-r components.
- **Unit–hypersphere normalization.** Before forming similarities/covariances, project embeddings onto the unit sphere, matching the contrastive geometry.
- **Symmetric case (unimodal).** When a unimodal subproblem reduces to estimating a symmetric target (e.g. solving $F^\top F$ in a single modality), use the symmetrized and ridge–shifted matrixand then apply eigendecomposition.

On the large–scale MSCOCO retrieval benchmark, we compare a baseline that uses a plain truncated SVD on $C(\gamma)$ against our stabilized pipeline above. The latter yields higher recall with negligible overhead.

Table 5: **Effect of stabilization on MSCOCO** (image–text retrieval). Stabilized SVD = regularization + randomized SVD (with power iterations) + unit-sphere normalization.

| SVD method | Train time (s) | R@1 | R@5 | R@10 |
|---|---|---|---|---|
| Standard truncated SVD | 32.47 | 0.2235 | 0.4486 | 0.5649 |
| **Stabilized SVD (UniCon)** | **33.28** | **0.2601** | **0.4990** | **0.6149** |

## C.5 EXPERIMENTS WITH OTHER MODALITIES

To further substantiate the general applicability of our approach with other complex modalities, we additionally evaluate UniCon on an audio text alignment task using the Clotho dataset(Drossos et al., 2019). In this experiment, we use pre-trained CLIP and Wav2CLIP(Wu et al., 2022) encoders to extract features from text and audio inputs respectively, followed by a linear projection layer for cross-modal alignment. The results show that without explicit alignment, the original feature structures exhibit a significant modality gap, while both UniCon and SGD achieve comparable and effective alignment performance after training. This additional experiment provides further evidence of our method's effectiveness in diverse modality alignment scenarios.

We believe this experiment with different modalities provide further support for UniCon's scalability and robustness.

Table 6: Audio-Text Alignment Results on the Clotho Dataset.

| Method | R@1 a2t | R@1 t2a | R@5 a2t | R@5 t2a | R@10 a2t | R@10 t2a | Time |
|---|---|---|---|---|---|---|---|
| No alignment | 0.0010 | 0.0000 | 0.0048 | 0.0029 | 0.0096 | 0.0096 | – |
| UniCon | 0.0335 | 0.0249 | **0.1311** | 0.1110 | **0.1943** | 0.1789 | **13.45s** |
| SGD-CLIP | **0.0373** | **0.0278** | 0.1244 | **0.1139** | 0.1923 | **0.2077** | 347.48s |

## C.6 KERNEL

**Why kernel?** The kernel-based formulation is essential in nonlinear contrastive alignment, as it enables an *implicit* mapping of feature representations into (potentially infinite-dimensional) Reproducing Kernel Hilbert Spaces (RKHS). This implicit lifting significantly enhances expressivity, allowing UniCon to capture complex cross-modal relationships that cannot be represented by linear projections alone, without explicitly constructing high-dimensional coordinates. Moreover, the kernel mapping effectively *unfolds* nonlinear manifolds (e.g., spherical or curved distributions) into a linearized feature space, where the spectral alignment mechanism can operate directly via rank-$r$ approximation.

**Why Angular Kernel?** We adopt angular kernels because features are normalized on the unit hypersphere, which is an effective practice in contrastive learning. Prior work(Wang et al., 2017) has shown that learning representations on the hypersphere leads to better performance than in Euclidean space, as it avoids the conflicting forces between attractive and repulsive gradients. Angular kernels are particularly well-suited for this geometry: they are theoretically sound, and simple to implement. In our view, this simplicity is not a limitation but an advantage. For comparison, we also experimented with the RBF kernel. The results confirmed our hypothesis: angular kernels consistently outperform RBF when embeddings lie on the hypersphere. It still worth to explore kernel approximation methods for memory efficient computation.

**Kernel Selection Study.** To further demonstrate the impact of kernel choice on alignment performance, we have included an empirical study in Table 7. The results below show the performance variance across different kernel types:

Table 7: Ablation study on kernel selection. Results demonstrate that kernels with stronger geometric expressivity (e.g., Angular and Arc-Cosine) yield superior alignment performance.

| Kernel Type | Synthetic Accuracy | CIFAR-10 Accuracy |
|---|---|---|
| RBF | .56 | .11 |
| Matérn | .73 | .44 |
| Cosine | .81 | .63 |
| Exponential Cosine | .73 | .63 |
| Arc-Cosine | .85 | .63 |
| Angular | **.86** | **.63** |

## C.7 LOSS VARIATIONS

**Support for Non-Smooth Losses (e.g., Triplet Loss).** Our generalized contrastive loss formulation accommodates both smooth and non-smooth cases. For losses such as the hinge-based triplet loss, classical gradients are not defined everywhere, yet their optimization is well-defined using *Clarke subgradients*. The rank-$r$ spectral characterization of UniCon still applies in this generalized subdifferential setting. To support this, we performed a synthetic nonlinear experiment (same setup as Sec. 4.1, replacing CLIP loss with triplet loss), where UniCon achieved **90% alignment accuracy**, confirming its compatibility with margin-based losses.

**Sigmoid-based Contrastive Losses.** We further evaluate UniCon under the Sigmoid contrastive loss used in SigLIP(Zhai et al., 2023). The results show that UniCon achieves performance comparable to SGD–SigLIP, demonstrating that UniCon is not limited to softmax-based contrastive objectives.

| Method | I2T R@1 | T2I R@1 | I2T R@5 | T2I R@5 | I2T R@10 | T2I R@10 |
|---|---|---|---|---|---|---|
| **UniCon (SigLIP)** | **0.3340** | **0.2862** | **0.5816** | 0.5334 | 0.6852 | 0.6394 |
| **SGD–SigLIP** | 0.2852 | 0.2816 | 0.5610 | **0.5538** | **0.6924** | **0.6704** |

Table 8: Comparison of UniCon and SGD–SigLIP under Sigmoid contrastive loss.

