# OpenReview forum: "UniCon: Unified Framework for Efficient Contrastive Alignment via Kernels"
_ICLR.cc/2026/Conference — ICLR 2026 Poster_

### Official Review · Reviewer_tNkS · 2025-10-28

**Soundness:** 4
**Presentation:** 3
**Contribution:** 4
**Rating:** 8
**Confidence:** 4

**Summary:**

This paper proposes UniCon (Unified Framework for Efficient Contrastive Alignment via Kernels), a theoretically grounded and computationally efficient alternative to stochastic contrastive learning. Unlike traditional contrastive methods such as CLIP or SimCLR that perform pairwise optimization between positive and negative samples through minibatch-based SGD, UniCon introduces a global contrastive similarity weight matrix $S(\gamma)$ that captures the pairwise interactions among all samples simultaneously.

## Technical Approach

UniCon formulates the contrastive objective in matrix form $C(\gamma) = X S(\gamma) Y^\top$ (or its kernelized version $M = K_X^{1/2} S(\gamma) K_Y^{1/2}$) and converts contrastive alignment into a spectral problem. Through rigorous derivations, the paper proves that minimizing the contrastive loss is equivalent to a rank-$r$ spectral approximation of $C(\gamma)$ in either linear or reproducing-kernel Hilbert space (RKHS) settings, yielding a closed-form global solution via singular value decomposition (SVD)—replacing thousands of gradient steps with a single spectral update. The authors further extend this formulation to nonlinear and many-to-many alignments through kernelization, demonstrating theoretical unification across linear, nonlinear, unimodal, and multimodal regimes.

## Empirical Results

UniCon achieves substantial speedups (100–400× faster) across synthetic and real datasets while maintaining or surpassing the performance of traditional SGD-based contrastive methods.

## Overall Impact

UniCon reframes contrastive learning as global spectral alignment rather than local pairwise optimization, providing both a unified theoretical interpretation and a dramatically accelerated training paradigm.

**Strengths:**

## Conceptual Novelty and Breakthrough

This paper presents a genuinely novel and impactful perspective on contrastive learning by reformulating it from a local, pairwise optimization problem into a global spectral alignment framework. The introduction of the contrastive similarity weight matrix $S(\gamma)$ and its associated closed-form spectral solution represents a conceptual breakthrough—showing that contrastive objectives, long thought to require large-batch SGD, can instead be solved analytically through singular value decomposition.

## Theoretical Rigor and Practical Impact

The problem addressed is fundamental and timely, given the enormous training cost and inefficiency of existing multimodal models like CLIP. By providing a mathematically rigorous unification of linear and nonlinear (kernelized) encoders, UniCon bridges theoretical understanding and practical acceleration, delivering both interpretability and efficiency within one framework.

## Empirical Validation

Empirically, the results are striking: UniCon achieves the same or better alignment performance while being up to hundreds of times faster than SGD-based CLIP training, across both synthetic and real-world multimodal datasets. The experiments are carefully designed to validate each theoretical claim—showing consistent convergence of $S(\gamma)$, clear geometric interpretability, and strong zero-shot transfer.

## Overall Assessment

This paper stands out for its original idea, theoretical depth, and practical relevance. It redefines how we think about contrastive learning optimization, offering a theoretically elegant and computationally transformative solution.

**Weaknesses:**

The paper lacks validation on large-scale datasets (such as LAION-400M or Conceptual-12M) and complex modalities including multilingual and video data. This limits our understanding of whether UniCon's spectral alignment approach scales effectively to real-world production settings and remains efficient when dealing with high-dimensional, heterogeneous modal representations.

While the theoretical formulation of $C(\gamma) = X S(\gamma) Y^\top$ is elegant, the paper would benefit from a clearer algorithmic description or pseudocode detailing how $C(\gamma)$ (or its kernelized version) is computed, aggregated, and updated in practice. This would improve reproducibility and help practitioners understand the computational flow beyond the mathematical abstraction.

**Questions:**

## 1. Optimality Under End-to-End Training

For end-to-end trainable encoders (non-frozen backbones), does the optimality of the spectral closed-form solution still hold? If the encoder weights evolve during training, would $S(\gamma)$ need to be recomputed and re-factorized frequently to maintain optimal alignment?

## 2. Low-Frequency Update Motivation

Why do the authors choose to update the global $C(\gamma)$ or $S(\gamma)$ at a low frequency? Is this design mainly motivated by computational efficiency, numerical stability, or to preserve the consistency of the learned spectral subspace as the encoder outputs evolve?

## 3. Distributional Shift Robustness

When the encoder undergoes significant distributional shifts (e.g., during fine-tuning or domain adaptation), could the low-frequency update scheme cause a mismatch between $S(\gamma)$ and the current feature distribution?

---

> ### Author Response · Authors · 2025-11-22
>
> We sincerely thank the reviewer for the highly positive assessment and constructive feedback.
>
> ### **W1: Scalability and Modality Coverage**
>
> - **Core Contribution and Experimental Design.**
> We would like to emphasize that the core contribution of this paper lies in **establishing a unified theoretical framework**. Accordingly, our experiments are designed to validate theoretical analysis, across unimodal, multimodal, and many-to-many cases with different scales.
>
> - **Why UniCon Can Generalize to Large-Scale Settings?**
>     - We sincerely acknowledge that experimenting at the scale of LAION-400M or Conceptual-12M is currently beyond our computational capabilities. However, we provide the following analysis and empirical results:
>     - Theoretically, contrastive alignment is fundamentally a rank-r spectral structure discovery problem (Section 3), which gives us intuition that we don't need massive datasets to find the principal axes.
>     - Empirically, on MSCOCO, we observe that using only 200 images (0.24\% of the dataset), (with each image paired with 5 captions), already yields 66.45\% avg R@10, indicating **data efficiency**.
>     - This suggests that UniCon has potential to be used as a warm-start initializer for large-scale contrastive models, reducing both optimization time and data requirements (Section 5, Appendix C3).
>
> - **Audio–Text Alignment.**
>
> To further substantiate the general applicability of our approach with other modalities, we have additionally evaluated UniCon on an audio-text alignment task using the Clotho dataset. In this experiment, we used pre-trained CLIP and Wav2Clip encoders to extract features from text and audio inputs respectively, followed by a linear projection layer for cross-modal alignment. The results show that without explicit alignment, the original feature structures exhibit significant modality gap, while both UniCon and SGD achieve comparable alignment performance after training. This additional experiment provides further evidence of our method's effectiveness in diverse modality alignment scenarios.
>
> | Method| R@1 a2t| R@1 t2a| R@5 a2t | R@5 t2a |R@10 a2t |R@10 t2a|time
> |-|-|-|-|-|-|-|-|
> |No alignment|0.0010|0.0000|0.0048|0.0029|0.0096|0.0096|0s
> |Unicon|0.0335|0.0249|**0.1311**|0.1110|**0.1943**|0.1789|**13.45s**|
> |SGD-CLIP|**0.0373**|**0.0278**|0.1244|**0.1139**|0.1923|**0.2077**|347.48s|
>
> We believe these experiments across different data scales and modalities provide substantial support for UniCon's theoretical soundness.
>
> ### **W2: Clarification on Algorithmic Description**
> We thank the reviewer for emphasizing the need for clearer algorithmic illustration.
> - The ready-to-implement code detailing the computation of $S(\gamma)$ is already provided in Appendix C.1 of our paper. Specifically, the exact form of $C(\gamma)$ and the kernalized version $M$ are derived from $S(\gamma)$ via the operations $C(\gamma)=XS(\gamma)Y^{\top}$ and $M=K_X^{1/2}S(\gamma)K_Y^{1/2}$ respectively.
>
> - In response to the reviewer's feedback, we **added pseudocodes in the Appendix C.1 Algorithm 1 and Algorithm 2** to illustrate the complete training pipeline of UniCon. These additions make the method straightforward to implement and clearly bridge the mathematical formulation with practical computation.
>
> ---
> [Clotho] K. Drossos, S. Lipping and T. Virtanen, "Clotho: an Audio Captioning Dataset," IEEE International Conference on Acoustics, Speech and Signal Processing (ICASSP), Barcelona, Spain, 2020, pp. 736-740, doi: 10.1109/ICASSP40776.2020.9052990.
>
> [Wav2clip] Wu, H. H., Seetharaman, P., Kumar, K., & Bello, J. P. (2022, May). Wav2clip: Learning robust audio representations from clip. In ICASSP 2022-2022 IEEE International Conference on Acoustics, Speech and Signal Processing (ICASSP) (pp. 4563-4567). IEEE.

---

> ### Author Response · Authors · 2025-11-22
>
> ### **Q1:**
>
> We appreciate this question. We clarify that our theoretical optimality results with r-rank aproximation is derived under the assumption that the input space of UniCon is static. It includes 2 cases:
> - Input space is data space (raw modalities). UniCon directly maps inputs to a shared rank-r space using kernelized operators. In this case, UniCon itself performs end-to-end alignment, and the spectral solution remains optimal.
> - Input space is embedding space from frozen encoders. This is the common setting in contrastive alignment (e.g., adapter training).
> Either case UniCon gives a global optimal spectral solution to contrastive loss minimization from the perspective of r-rank approximation.
>
> When encoders are trainable (non-static input space), UniCon is applied during jointly optimizing encoders, the spectral update becomes a conditionally optimal subproblem, i.e., optimal for the current encoder outputs. In practice, one can only recompute once per meta-epoch rather than each step. This motivates a hybrid spectral–SGD strategy or warm-start strategy, and we discuss this potential future direction in Appendix C5.
>
>
>
> ### **Q2:**
> - We update $S(\gamma)$ not very frequently first becuase it becomes stable very fast for a fixed input space, either direct data space or embedding space from a frozon backbone. Refer to Algorithm discussion in Appendex C.1, it is the theoretical basis of this framework that UniCon achieves a spectral update with the same performance of many gradient steps.
>
> - In addition, we note that UniCon is data efficient (See the reply to weakness 1). From this perspective, we can also update the global $S(\gamma)$ at a low frequency without degrading performance.
>
>
> ### **Q3:**
> - In Table 2, we have implemented zero shot transfer on Flicker30k and MSCOCO, which demonstrates that $S(\gamma)$ learned from UniCon is robust under moderate distributional shift.
>
> - When encoder features experience large distributional shifts during fine-tuning, one can do few-shot learning to fit the current domain. Because UniCon yields a global spectral solution, such recomputation is computationally efficient compared to per-step SGD updates; and data efficiency property as in response of W1 enables few-shot.
>
>
> We are grateful for the reviewer’s generous evaluation, particularly regarding conceptual novelty and theoretical rigor. We believe that UniCon offers a unified spectral interpretation of contrastive learning covering linear and nonlinear settings, one-to-one and many-to-many alignment. We welcome further discussions.

---

> > ### Comment · Reviewer_tNkS · 2025-11-26
> >
> > Thank you for the detailed clarifications.

---

### Official Review · Reviewer_WMSJ · 2025-10-30

**Soundness:** 2
**Presentation:** 2
**Contribution:** 2
**Rating:** 2
**Confidence:** 4

**Summary:**

To address the slow training problem of the state-of-the-art multimodal models, the paper proposes a Unified Framework for Efficient Contrastive Alignment via Kernels (UniCon). Specifically, it uses the contrastive similarity weight matrix to find the closed-form global solutions, instead of minibatch backpropagation.

**Strengths:**

1. UniCon provides a kernel-based perspective analysis showing the connection between contrastive loss and the spectral update.

2. UniCon shows faster convergence than the CLIP-SGD.

**Weaknesses:**

1. In Lemma 4, it states $\beta=\beta(\theta_1,\theta_2)$. What is $\beta$ here? It is not explained, and $\beta$ does not appear in the equation. Please clarify (likely $\gamma$).

2. Typo (line 729): “Equation equation 2” → “Equation 2”.

3. In line 134 you write “scaling factor $\nu \ge 1$,” while line 148 says “$\nu > 0$.” For rigor, please use a consistent and correct domain for $\nu$.

4. $\gamma$ depends on $\phi'$ and $\psi'$ (Def.~3), but choices like the triplet loss that is mentioned multiple times are non-smooth.


5. The statement “UniCon is the first implementation that directly leverages the contrastive similarity weight $S(\gamma)$ for contrastive alignment” is too strong. For example, [1] leverages similarity weights via optimal transport, and [2] uses kernel-based alignment for multimodal contrastive learning.


6. Figure 2 is also not clear to me. The two T-SNE figures for SGD-CLIP and UniCon look identical to me. What benefit beyond training time is demonstrated? Please clarify or add quantitative metrics.

7. Since efficiency is a claimed contribution, a convergence analysis (or at least a theoretical discussion) is needed to justify why the method should be more efficient.

8. Experiments are limited to small projection heads (e.g., two-layer MLPs). Real multimodal setups are larger. It is unclear how the method scales to larger/complex heads, partial/full CLIP fine-tuning, or training from scratch. Some baselines appear underpowered; e.g., [3] shows that with appropriate adapters/architectures, SGD-CLIP can perform well (including zero-shot). Given the abstract’s opening claim about slow training of state-of-the-art models, it would be fair to evaluate efficiency at that scale; current comparisons are not yet convincing.


[1]. Shi L, Fan J, Yan J. Ot-clip: Understanding and generalizing clip via optimal transport[C]//Forty-first International Conference on Machine Learning. 2024.

[2]. Gong S, Jiang Y, Dou Q, et al. Kernel-based unsupervised embedding alignment for enhanced visual representation in vision-language models[J]. ICML, 2025.

[3]. Gao P, Geng S, Zhang R, et al. Clip-adapter: Better vision-language models with feature adapters[J]. International Journal of Computer Vision, 2024, 132(2): 581-595.

**Questions:**

1. The paper claims faster convergence than the CLIP objective, but no convergence curves are shown. Please add plots comparison (e.g., epoch/time vs. validation accuracy) and report CLIP results with other optimizers (SGD/Adam/AdamW). How were hyperparameters (optimizer, learning rate, etc.) tuned for CLIP and for your method? Is the comparison fair?

---

> ### Author Response · Authors · 2025-11-22
>
> We thank the reviewer for insightful comments and for taking the time to review our work.
>
> ### **Novelty (W5)**
> We thank the reviewer for the feedback. We have revised the phrasing in Section 4 line 326-328 to be more modest, and added citations on line 49 and line 118. However, we would like to clarify that our use of contrastive similarity weight matrix and kernels are foundamentally different from [1] and [2]:
>
> - Relation to OT-CLIP[1]: **Did we use the same "similarity weight matrix"?**
> No. The similarity weight matrix in UniCon defined in Definition 3 is constructed using $\phi,\psi$ and derived from minimizing our proposed generalized contrastive loss (see Appendix B). This construct is not present in OT-CLIP.
> OT-CLIP provides an interpretation of the standard CLIP loss by framing it as an OT-based bilevel optimization problem. These are complementary, insightful, but different contributions based on different premises.
>
> - Relation to [2]: **Are kernels used in the same way?**
> No. While [2] also incorporates kernels, its use of kernel is fundamentally different from ours in motivation, task, and methodology. Specifically, [2] matches Gram matrices to refine CLIP's visual features toward those of a stronger model (DINO) in single modality.
> In contrast, UniCon introduces kernels as intrinsic representation operators in Reproducing Kernel Hilbert Space (RKHS), enabling the extension of the linear spectral theory into nonlinear settings, and leading to a general kernelized contrastive alignment theorem (Theorem 9).
>
> - **Clarified contribution**.
> Our contribution is not simply “using similarity weight matrix” or “using kernels,” but in:
>     1. **Unified theoretical framework to understand contrastive alignment.**
> We show that minimizing contrastive loss is essentially performing a rank-r spectral approximation with RKHS. This provides a unified framework in which linear and nonlinear encoders for contrastive alignment are treated under the same $r$ -rank spectral view.
>     2. **From one-to-one to many-to-many alignment.**
> We formulate a general family of contrastive losses that covers many-to-many alignment (e.g., multiple captions per image), broadening the applicability of contrastive alignment.
>     3. Gradient-induced $S(\gamma)$.
> Starting from this generalized loss family, we derive the contrastive similarity weight matrix $S(\gamma)$, which yields a concise and unified solution applicable to both the linear case (via $C(\gamma)$) and in the nonlinear settings (via the kernelized operator $M$).
>
> ### **Efficiency (W7 and Q1)**
> We thank the reviewer for raising this point. We acknowledge that our use of the term *convergence* may have been ambiguous, and we will clarify this more explicitly.
> In UniCon, our efficiency claim refers to the rapid stabilization of the alignment subspace through derived spectral updates, which bypass many small gradient steps. Details can be found in Algorithm Discussion in Appendix C1.
>
>     Why is this faster?
>
> - We provide an intuitive explanation: Unlike gradient-based methods that take small local steps, each spectral update directly jumps to the global maximizer of the surrogate objective, making the update much more informative.
> - Empirically, we observe an interesting phenomenon that the $M$ (or $C(\gamma)$ in linear case) converge in 2 or few steps.
> - Additionally, on MSCOCO, using only 200 images (0.24% of the dataset), with each image paired with 5 captions, already yields meaningful retrieval alignment, indicating both subspace convergence and potential **data efficiency**, enabled by one-to-many alignment insights of UniCon.
> - A full theoretical characterization of convergence rates and generalization bounds is beyond the current scope, but we provide:
>     1. Empirical evidence of rapid **wall-clock efficiency**, supporting the claimed practical efficiency.
>     2. Insights into how our method may serve as a warm-start or alignment initialization for large-scale training.
>
> ### **Clarification on Figure 2 (W6)**
> We clarify that the goal of Figure 2 is **not to show a visual difference** between SGD-CLIP and UniCon, but rather to show that **UniCon achieves comparable alignment quality** with SGD-CLIP using only a few spectral steps, without gradient-based training.
>
> The visual similarity between the two t-SNE plots is in fact **expected and desired**: UniCon is designed to match the representational quality of SGD-CLIP, not to produce a different structure. Both methods successfully map two modalities (cross/circle) into a shared latent space where:
> - matched pairs are closely aligned (connected by lines),
> - cluster structure is preserved across modalities (color grouping),
> - inter-modal geometry is comparable.
>
> To make this clearer, we revised the figure captions to explicitly state this intended equivalence.

---

> ### Author Response · Authors · 2025-11-22
>
> ### **Projection Head Complexity (W8)**
> We appreciate this comment. We provide a unified theoretical framework for contrastive alignment, rather than propose a new architecture. This allows UniCon to conceptually help understand contrastive model with linear and nonlinear projectors. Accordingly, our experiments are designed to validate the theoretical claims.
> - Thank you for pointing out the relevant work [3]. We additionally tested CLIP-Adapter–style heads (MLP + residual adapter) and observed marginal performance improvement (less than 1%). One possible explanation is that CLIP-Adapter is designed for classification under vision-language cross-modal settings, while our experiments on CIFAR10 focuses on unimodal classification leveraging contrastive alignment. We follow the standard SimCLR setup with a widely adopted 2-layer MLP (FC–BN–ReLU–FC).
> - We will add a brief discussion on how UniCon can serve as a fast warm-start initializer for large-scale multimodal models in Appendix C5.
>
> ### **Optimizer Fairness (Q1)**
> To ensure a fair comparison, we conducted a comprehensive study of various optimizers (including SGD and Adam and AdamW), evaluating different configurations of momentum and weight decay. The results reported in our paper represent the best-performing configurations (AdamW, lr=1e−3) identified through this process, as explicitly described in Section 4.1.Below is a brief summary of the optimizer study:
>
> | Setting              | Optimizer | Accuracy | Time (s) | Notes |
> |----------------------|-----------|--------------------:|---------:|-------|
> | Synthetic Linear     | AdamW     | **1.00**            | 0.53     | Fully converged |
> |                      | SGD       | 0.92                | 0.55     | Slightly slower|
> | Synthetic Nonlinear  | AdamW     | **0.84**            | 0.65     |       |
> |                      | SGD       | 0.85                | **21.00**|       |
> | Flickr30k     | AdamW     | R@1: 0.236, R@5: 0.597  | **45.3** | Best baseline |
> |                      | SGD       | R@1: 0.1856, R@5: 0.530         | 2386.11  | Did not converge in 2500 epochs |
>
>
> To further demonstrate fairness, in Appendix C2, we've already provided loss plots or accuracy plots for the experiments in the paper, which shows that all the gradient based iterative method in the paper are fully trained to converge.
>
> ### **Other comments (W1-W4)**
> - Thank you for suggested edits, including the feedback on the supplementary material. The typos have been corrected in the updated version.
> - Regarding non-smooth losses (e.g., triplet loss):
>     - Our generalized contrastive loss (Definition 2) supports nonsmooth cases through Clarke subgradients. The spectral characterization still holds under subdifferentiability.
>     - Added empirical results: A quick check of validity can be conducted on synthetic nonlinear data (setup is the same as in section 4.1 except replacing CLIP loss with triplet loss setting), which achieves 90% accuracy.
>
> Thank you again for your helpful feedback and response. We welcome any additional discussions.

---

### Official Review · Reviewer_5pMu · 2025-10-31

**Soundness:** 2
**Presentation:** 3
**Contribution:** 3
**Rating:** 6
**Confidence:** 2

**Summary:**

The paper proposes UniCon, a kernel-based framework for efficient contrastive alignment that yields closed-form spectral updates instead of gradient-based training. The method introduces a contrastive similarity weight matrix and shows that minimizing a broad family of contrastive losses is equivalent to maximizing a trace objective whose optimizer is characterized by a spectral decomposition, both for linear encoders and for nonlinear encoders via RKHS kernels.

**Strengths:**

(1) Motivation and scope are clear.
(2) Unified kernelized theory and shows gradient equivalence from general contrastive losses to a trace objective, extends linear SVD view to RKHS with an explicit optimizer.
(3) The method achieves outstanding performance with synthetic-linear scaling and 100% matching in 0.02 s (compared to 0.32 s for SGD), reaching similar accuracy in ~2× less time and with only ~2 epochs on the CIFAR-10 dataset.

**Weaknesses:**

(1) The kernel selection and the corresponding ablation are not explained in detail. The paper recommends an angular kernel and notes it outperforms RBF when embeddings are on the hypersphere but quantitative kernel ablations are not shown beyond a qualitative statement.
(2) Limited experimental scope. All experiments use frozen or very shallow trainable encoders and comparison baselines are limited to vanilla SGD-CLIP without comparing against other efficient contrastive learning methods.
(3) Theorem 9 contains a typo in the title ("characterization") and I think the author may optimize the notation in the article.

**Questions:**

Please see weaknesses.

---

> ### Author Response · Authors · 2025-11-22
>
> We thank the reviewer for the constructive feedback and positive assessment. Below we address each weakness in turn.
>
> ### **Kernel Selection**
> We agree that our original explanation of kernel selection was too qualitative. In the revised manuscript (Appendix C.7), we now include a detailed ablation that quantitatively compares kernel choices on both synthetic and CIFAR-10 datasets:
>
> | Kernel Type                     | Synthetic Accuracy | CIFAR-10 Accuracy |
> |----------------------------------|--------------------:|------------------:|
> | RBF Kernel                      | 0.56               | 0.11            |
> | Matérn Kernel                  | 0.73               | 0.44            |
> | Cosine Kernel                  | 0.81               | 0.63            |
> | Exponential Cosine Kernel      | 0.73               | 0.63            |
> | Arc-Cosine Kernel  | 0.85           | 0.63            |
> | Angular Kernel                 | **0.86**                | **0.63**        |
>
> - Kernels that respect **hyperspherical geometry** (angular, arc-cosine, cosine) consistently outperform distance-based kernels like RBF, especially when encoder outputs are normalized, which is common in CLIP/SimCLR setups.
> - As mentioned in the Section 5, how random Fourier features and learnable kernels can be adopted to scale beyond above kernels, marking a promising future direction.
>
> ### **Limited Experimental Scope**
>
> This work primarily contributes a unified theoretical framework for contrastive alignment, rather than proposing a new architecture. UniCon offers another perspective for analyzing contrastive alignment. Accordingly, our experiments are designed to validate theoretical analysis, across unimodal, multimodal, and many-to-many cases. Since the framework is formally equivalent to minimizing the contrastive loss, we compare against traditional gradient-based baselines, such as minimizing CLIP loss by stochastic gradient descent (SGD-CLIP), to fairly and directly substantiate our claims.
>
> From this perspective, to address the concern:
> - Loss selection: We added experiments under Sigmoid contrastive loss in SigLIP (Zhai et al., 2023) and observed comparable performance than SGD-SigLIP, confirming that UniCon is not limited to one loss family.
>
> | Method           | I2T R@1 | T2I R@1 | I2T R@5 | T2I R@5 | I2T R@10 | T2I R@10 |
> |------------------|--------:|--------:|--------:|--------:|---------:|---------:|
> | **UniCon (SigLIP)** | **0.3340** | **0.2862** | **0.5816** | 0.5334 | 0.6852 | 0.6394 |
> | **SGD–SigLIP**     | 0.2852 | 0.2816 | 0.5610 | **0.5538** | **0.6924** | **0.6704** |
>
> - Projector design: We additionally tested CLIP-Adapter–style (Gao et al., 2024) heads (MLP + residual adapter) and observed marginal performance improvement (less than 1%). One possible explanation is that CLIP-Adapter is designed for classification under vision-language cross-modal settings, while our experiments on CIFAR10 focuses on unimodal classification leveraging contrastive alignment. We follow the standard SimCLR setup with a widely adopted 2-layer MLP (FC–BN–ReLU–FC).
>
> - Optimizer selection: We conducted a comprehensive study of various optimizers (including SGD and AdamW), evaluating different configurations of momentum and weight decay. The results reported in our paper represent the best-performing configurations (e.g. AdamW, lr=1e−3 in section 4.1) identified through this process.
>
> - In addition, UniCon has potential to serve as a fast warm-start initializer for large-scale multimodal models, which is discussed in Appendix C5.
>
> ### **Other Comments**
> Typo in Theorem 9: Thank you for pointing this out, and we have corrected.
>
>
> We emphasize that UniCon reveals that contrastive learning is essentially a **r -rank spectral approximation** problem with RKHS, **unifying** linear, nonlinear, one-to-one, and *many-to-many* contrastive objectives under the same framework.
>
> We sincerely appreciate the reviewer’s insightful comments. We welcome additional feedback or clarification.
>
> ---
> Gao, P., Geng, S., Zhang, R., Ma, T., Fang, R., Zhang, Y., ... & Qiao, Y. (2024). Clip-adapter: Better vision-language models with feature adapters. International Journal of Computer Vision, 132(2), 581-595.
>
> Zhai, X., Mustafa, B., Kolesnikov, A., & Beyer, L. (2023). Sigmoid loss for language image pre-training. In Proceedings of the IEEE/CVF international conference on computer vision (pp. 11975-11986).

---

### Official Review · Reviewer_GRPF · 2025-11-01

**Soundness:** 3
**Presentation:** 3
**Contribution:** 3
**Rating:** 6
**Confidence:** 2

**Summary:**

This paper presents UniCon, a novel and theoretically grounded framework that reformulates contrastive learning as a spectral decomposition problem. The core contribution is the introduction of a contrastive similarity weight matrix, which allows for the replacement of iterative, gradient-based optimization with a closed-form spectral update. The framework unifies linear and nonlinear (via RKHS kernelization) encoders and extends from one-to-one to many-to-many alignment settings. The empirical results across synthetic, unimodal, and multimodal tasks are compelling, demonstrating performance competitive with or superior to SGD-based baselines.

**Strengths:**

- Theoretical Unification and Novelty
- Comprehensive Empirical Validation
- The paper is generally well-structured, with a clear road-map of the theoretical contributions.

**Weaknesses:**

- Breadth of Baseline Comparison: The paper mainly compares to SGD-CLIP, without other recent strong baselines.
- The paper provides a novel kernel implementation, but provides limited ablation or intuition as to why.
- The introduction provides limited hints about the shortcomings of previous works and why the current study is necessary.
- The introduction claims that the linear and nonlinear settings are unified. However, this claim is questionable since the linear and nonlinear methodologies are almost always stated separately.
- The superiority of the nonlinear setting compared to the linear one needs to be stated clearly, e.g., by providing an intuitive example.
- The Experiment section lacks the direct comparison of performances between linear and nonlinear methods, either by Figures or quantitative results (Table 1 and 2).
- The visualization difference between UniCon and SGD-CLIP (Figure 2 (mid vs right), Figure 4 (b) vs (c)). Consider adjusting the visualization to make the difference more evident.
- The Discussion section simply provides some observations and conclusions, which makes it difficult to differentiate from the Conclusion. Is it possible to add some in-depth analysis that supports these observations?

**Questions:**

- Add more baselines other than SGD-CLIP.
- Provide more comparative results between linear and nonlinear; between UniCon and SGD-CLIP.
- Add more in-depth analysis.
- Add a detailed example to illustrate why the nonlinear setting is better than linear.

---

> ### Author Response · Authors · 2025-11-22
>
> We thank the reviewer identify our work as a novel and theoretically grounded framework.
>
> ### **Novelty and Motivation of the Unified Framework**
>
>     Why the current study is necessary?
>
> 1. **From linear theory to practical nonlinear multimodal models**
>
> Prior theoretical work (e.g., Nakada et al., 2023) provides an elegant analysis of multimodal contrastive learning **under linear projection assumptions**. However, this linearity assumption is overly restrictive and fails to reflect real-world practices:
>
>    - In SimCLR, Chen et al. (2020) show that a nonlinear projector significantly outperforms a linear one.
>    - BYOL (Grill, J.-B. et al., 2024) and many subsequent self-supervised methods adopt similar nonlinear MLP heads.
>    - As summarized in Li & Tang (2024), many recent multimodal alignment frameworks employ multi-layer perceptrons (MLPs) or other nonlinear projectors after pretrained encoders to align heterogeneous feature spaces across modalities.
>
> These observations indicate that a theory restricted to linear projectors misses the behavior of the models actually used in practice. Our work is necessary to provide a unified framework that covers both linear and nonlinear (kernelized) projection heads, extending the spectral viewpoint to RKHS.
>
> 2. **From one-to-one to many-to-many alignment**
>
> Existing analyses mostly assume **one-to-one** positive pairs, which is not always true in practice. For instance, in our experiments on MS-COCO, a single image is associated with five captions, all semantically valid positives.
>
> Our framework explicitly incorporates many-to-many positive relationships into the analysis and into the construction of the similarity weight matrix. This broadens the applicability of spectral contrastive alignment.
>
>     Why "unified" for linear and nonlinear? The introduction claims that the linear and nonlinear settings are unified. However, this claim is questionable since the linear and nonlinear methodologies are almost always stated separately.
>
> We thank the reviewer for this insightful feedback. We have now clarified this in Section 3.3, showing rigorously that the nonlinear (kernel) formulation strictly contains the linear setting as a special case.For clarity, we restate the key point here:
>
> - **Linear case is recovered exactly when using linear kernels.**
> For linear kernels $k_X(x,x')=\langle x,x'\rangle$, $k_Y(y,y')=\langle y,y'\rangle$, the Gram matrices reduce to $K_X=X^{\top}X$ and $K_Y=Y^{\top}Y$. The kernel alignment operator in our nonlinear formulation $$M=K_X^{1/2}S(\gamma)K_Y^{1/2}$$ becomes $$M=(X^{\top}X)^{1/2}S(\gamma)(Y^{\top}Y)^{1/2}.$$
> **The top-r spectral structure of M is mathematically equivalent (up to orthogonal rotations) to the weighted contrastive covariance $C(\gamma)=XS(\gamma)Y^{\top}.$** Therefore, the kernelized setting provides a superset of the linear case.
>
> - **Unified interpretation.**
>
> Theorem 9 provides a unified spectral view for understanding contrastive alignment: **contrastive loss minimization $\Longleftrightarrow$best rank–$r$ approximation with RKHS.**
>
>     The superiority of the nonlinear setting compared to the linear one needs to be stated clearly, e.g., by providing an intuitive example.
>
> In begin of section 3.2, we provided a brief motivation for why theoretical analysis should leave linear setting towards nonlinear setting. Here we add more explanation:
>
> - Intuitively, the linear UniCon setting can only discover a global linear subspace that aligns the two modalities. This is sufficient when the cross-modal relationship is approximately linear (e.g., when one modality is an affine transformation of the other). However, **in realistic problems, the alignment is often highly nonlinear in feature space.**
>
> - **Illustrative example**. Consider a simple 2D input space where image embeddings lie on a circle (points arranged by color) and the corresponding text embeddings lie along a line (e.g., ranking captions from “dark red object” to “bright yellow object”). A linear projector can only align these spaces by flattening the circle into a line, which inevitably breaking neighborhoods. In contrast, the nonlinear formulation with RKHS mapping makes circle manifold is unfolded into a high-dimensional feature space, so alignment preserves both local neighborhoods and global semantic structure.
>
> - Nonlinear UniCon strictly generalizes linear UniCon (Section 3.3).Therefore,
>     - Anything learnable by linear UniCon is also learnable by nonlinear UniCon,
>     - but nonlinear UniCon can additionally capture complex nonlinear relationships.

---

> ### Author Response · Authors · 2025-11-22
>
> ### **Why kernel for nonlinear setting?**
> - As discussed in Wainwright (2019), RKHS captures **a rich class of functions** with favorable statistical and computational properties. This allows UniCon to model contrastive alignment beyond linear subspace assumptions.
> - The kernel approach provides a principled mathematical framework that enables implicit mapping of features into high-dimensional reproducing kernel Hilbert spaces. This mechanism significantly **enhances expressivity**, capturing intricate feature relationships without explicitly constructing high-dimensional coordinates.
> - Importantly, kernelization allows us to **retain the same spectral alignment interpretation**,i.e., contrastive loss minimization is essentially best rank-r approximation, now extended to RKHS. The linear case becomes a special instance.
> - To substantiate this claim, we have included both an illustrative example in our previous response and a comprehensive ablation study on kernel selection in Appendix C.7 in the revised manuscript.
> - Looking forward, we recognize the potential of more sophisticated kernel designs. In particular, exploring random Fourier features represents a promising direction for future work. (mentioned in section 5.)
>
> ### **Breadth of Baseline Comparison**
>
> This work primarily contributes a unified theoretical framework for contrastive alignment, rather than proposing a new architecture. UniCon offers another perspective for analyzing contrastive alignment under both linear and nonlinear projectors. Accordingly, our experiments are designed to validate theoretical analysis rather than pursue state-of-the-art performance. Since the framework is formally equivalent to minimizing the contrastive loss, we compare against traditional gradient-based baselines, such as minimizing CLIP loss by stochastic gradient descent (SGD-CLIP), to fairly and directly substantiate our claims.
>
> From this perspective, we address the question of "stronger baselines" along three aspects:
> - Projector design: We additionally tested CLIP-Adapter–style (Gao et al., 2024) heads (MLP + residual adapter) and observed marginal performance improvement (less than 1%). One possible explanation is that CLIP-Adapter is designed for classification under vision-language cross-modal settings, while our experiments on CIFAR10 focuses on unimodal classification leveraging contrastive alignment. We follow the standard SimCLR setup with a widely adopted 2-layer MLP (FC–BN–ReLU–FC).
> - Optimizer selection: We conducted a comprehensive study of various optimizers (including SGD and AdamW), evaluating different configurations of momentum and weight decay. The results reported in our paper represent the best-performing configurations (AdamW, lr=1e−3) identified through this process, as explicitly described in Section 4.1.
> - Loss selection: We added experiments under Sigmoid contrastive loss in SigLIP (Zhai et al., 2023) and observed comparable performance than SGD-SigLIP, confirming that UniCon is not limited to one loss family.
> | Method           | I2T R@1 | T2I R@1 | I2T R@5 | T2I R@5 | I2T R@10 | T2I R@10 |
> |------------------|--------:|--------:|--------:|--------:|---------:|---------:|
> | **UniCon (SigLIP)** | **0.3340** | **0.2862** | **0.5816** | 0.5334 | 0.6852 | 0.6394 |
> | **SGD–SigLIP**     | 0.2852 | 0.2816 | 0.5610 | **0.5538** | **0.6924** | **0.6704** |
>
> ### **Clarification on Figure 2 and 4**
> We clarify that the goal of Figure 2 is **not to show a visual difference** between SGD-CLIP and UniCon, but rather to show that **UniCon achieves comparable alignment quality** with SGD-CLIP using only a few spectral steps, without gradient-based training.
>
> The visual similarity between the two t-SNE plots is in fact **desired**: UniCon is designed to match the representational quality of SGD-CLIP, not to produce a different structure. Both methods successfully map two modalities (cross/circle) into a shared latent space where:
> - matched pairs are closely aligned (connected by lines),
> - cluster structure is preserved across modalities (color grouping),
> - inter-modal geometry is comparable.
>
> Similar explanations apply to Figure 4. To make this clearer, we revised the captions to explicitly state this intended equivalence.

---

> ### Author Response · Authors · 2025-11-22
>
> ### **Analysis on Linear–Nonlinear unification added in Section 3.3**
> We thank the reviewer for suggesting a more in-depth analysis. We added justification in Section 3.3, showing that the nonlinear (kernel) formulation strictly generalizes the linear case, recovering the linear format exactly when using linear kernels, thus establishing UniCon as a single unified spectral framework applicable to both linear and nonlinear contrastive alignment.
>
> - **Linear case is recovered exactly when using linear kernels.**
> For linear kernels $k_X(x,x')=\langle x,x'\rangle$, $k_Y(y,y')=\langle y,y'\rangle$, the Gram matrices reduce to $K_X=X^{\top}X$ and $K_Y=Y^{\top}Y$. The kernel alignment operator in our nonlinear formulation $$M=K_X^{1/2}\,S(\gamma)\,K_Y^{1/2}$$ becomes $$M=(X^{\top}X)^{1/2}S(\gamma)(Y^{\top}Y)^{1/2}.$$
> **The top-r spectral structure of M is mathematically equivalent (up to orthogonal rotations) to the weighted contrastive covariance $C(\gamma)=X\,S(\gamma)\,Y^{\top}.$** Therefore, the kernelized setting provides a superset of the linear case.
>
> - **Unified interpretation.**
>
> Theorem 9 provides a unified spectral view for understanding contrastive alignment: *contrastive loss minimization $\Longleftrightarrow$best rank–$r$ approximation with RKHS.*
>
> ### **More Analysis**
> We appreciate the reviewer’s suggestion to enhance the depth of Analysis. In response, apart from aforementioned parts, we added more analysis on:
> - Computational efficiency in Section 5 and details can be found in Algorithm Discussion in Appendix C.1.
> - Data Efficiency in Section 5.
> - Training strategies with static or evovling input spaces in Section 5 and Appendix C.5.
> - Experiments with other modalities in Appendix C.6.
> - Kernel selection study in Appendix C.7.
>
>
> Thank you again for helpful feedback, which helped improve the clarity and impact of the manuscript. We welcome any additional discussions.
>
> ---
> Nakada, Ryumei, et al. "Understanding multimodal contrastive learning and incorporating unpaired data." International Conference on Artificial Intelligence and Statistics. PMLR, 2023.
>
> Chen, T., Kornblith, S., Norouzi, M., & Hinton, G. (2020, November). A simple framework for contrastive learning of visual representations. In International conference on machine learning (pp. 1597-1607). PmLR.
>
> Grill, J. B., Strub, F., Altché, F., Tallec, C., Richemond, P., Buchatskaya, E., ... & Valko, M. (2020). Bootstrap your own latent-a new approach to self-supervised learning. Advances in neural information processing systems, 33, 21271-21284.
>
> Li, S., & Tang, H. (2024). Multimodal alignment and fusion: A survey. arXiv preprint arXiv:2411.17040.
>
> Wainwright, M. J. (2019). High-dimensional statistics: A non-asymptotic viewpoint (Vol. 48). Cambridge university press.
>
> Gao, P., Geng, S., Zhang, R., Ma, T., Fang, R., Zhang, Y., ... & Qiao, Y. (2024). Clip-adapter: Better vision-language models with feature adapters. International Journal of Computer Vision, 132(2), 581-595.
>
> Zhai, X., Mustafa, B., Kolesnikov, A., & Beyer, L. (2023). Sigmoid loss for language image pre-training. In Proceedings of the IEEE/CVF international conference on computer vision (pp. 11975-11986).

---

### Author Response · Authors · 2025-12-03
**Global Response**

We would like to thank reviewers for their constructive feedback. We are pleased to see that our work is recognized by reviewers from the perspective of motivation, theoretical novelty, empirical validation, and clear presentation.

Comments such as
> "Theoretical unification and novelty." (reviewer GRPF)
>
> "It redefines how we think about contrastive learning optimization, offering a theoretically elegant and computationally transformative solution." (reviewer tNkS)
>
> "Comprehensive empirical validation." (reviewer GRPF)
>
> "Motivation and scope are clear." (reviewer 5pMu)

reflect the core strengths we intended to contribute.

We highlight that most comments have been addressed in the updated manuscript available on OpenReview. Below, we’ve compiled rebuttal points asked by several reviewers while specific questions are answered in the individual rebuttals.


1. **Why UniCon is Necessary**:
   - **From linear to nonlinear**: Prior theoretical frameworks analyze contrastive learning mainly under linear projection assumptions, which do not explain real-world setups that employ nonlinear projectors. UniCon unifies both linear and nonlinear via RKHS.
   - **From one-to-one to many-to-many**: We formalize many-to-many alignment (e.g., multiple captions per image, class-level positives) in the generalized contrastive loss, handled by the derived $S(\gamma)$.

    With above unification, UniCon shows that contrastive loss minimization is equivalant to rank-r spectral approximation with RKHS.

2. **Unification of linear & nonlinear settings**:
    - We illustrate that linear case can be recovered from nonlinear case exactly using linear kernels. And the top $r$ spectral structure of $M$ in nonlinear case is mathematically equivalent (up to orthogonal rotations) to that of $C(\gamma)$ in linear case.
    - To sum up, the main theorem, which states that contrastive loss minimization is equivalent to best rank-r approximation with RKHS, is unified. We add a thorough clarification in Section 3.3.

4. **Experimental clarification and added studies**:
Our experiments are designed to validate the theoretical insights of UniCon. To support this, we added during rebuttal:

   - **Added experiments**: We add an **audio-text** retrieval task experiment to further substantiate the applicability of UniCon with other modalities. The results show that UniCon and SGD achieve comparable alignment performance. The detailed results can be found in Appendix C.6.
   - **Additional analysis**: Given the purpose of our experiments, we add additional analysis as follows.
       - Optimizer selection: We have evaluated different configurations of optimizer and the results reported represent the best-performing one.
       - Loss function: We added experiments using Sigmoid contrastive loss and observed comparable performance between UniCon and SGD-SigLIP in Appendix C.8.
       - Kernel selection: We conducted a detailed comparison between different kernel choices and observed that kernels respecting hyperspherical geometry consistently outperform distance-based kernels. The details can be found in Appendix C.7.

**Other manuscript improvements**:
- Sec. 3.3: Unified linear/nonlinear reduction (theory)
- Sec. 5 (Discussion): Discussion enriched with ovservations and anlysis of computational efficiency and data efficiency.
- Figure 2&4 captions updated: Clarified role of figures.
- App. C.1: algorithmic pseudocode
- App. C.5: Hybrid spectral-SGD training potential.

We welcome further discussion and thank the reviewers and AC again for their time.

---

### Meta-Review · Area_Chair_tkUq · 2026-01-10

**Summary:**

Reviews were mixed, with one reviewer giving an accept rating (8) with high conference (4), two reviewers (GRPF, 5pMu) providing borderline acceptance ratings (6) with low confidence (2), and one reviewer (Gms2) recommending rejection (2) with high confidence (4). Key weaknesses consistently raised were the initial lack of clarity in ablation studies, the perceived limited experimental scope (especially comparisons to other strong baselines beyond SGD-CLIP and scaling to very large datasets/architectures), and challenges in clarifying the linear/nonlinear unification.

**Reviewer Concerns:**

The authors provided a comprehensive and highly responsive rebuttal, including expanded ablation studies with quantitative results, offered theoretical proofs and illustrative examples to clarify the linear/nonlinear unification, and provided detailed runtime/memory analyses to address efficiency and scalability concerns. Furthermore, they added experiments with alternative loss functions (SigLIP) and an audio-text modality, demonstrating broader applicability, and clarified the paper's novelty by differentiating it from existing works leveraging kernels or similarity matrices.
These issues are well addressed.

**Reviewer Scores:**

Given the author's detailed response to the reviewers' questions, the three reviewers who initially gave positive scores (8,6,6) will maintain their ratings, while the reviewer who gave the lower score (2) may raise his score slightly.

---

### Decision · Program_Chairs · 2026-01-26

Accept (Poster)